# Feedback mechanisms stabilise degraded turf algal systems at a $CO_2$ seep site

Ben P. Harvey [1,5✉], Ro Allen [2,3,5], Sylvain Agostini[1], Linn J. Hoffmann[2], Koetsu Kon[1], Tina C. Summerfield[2], Shigeki Wada[1] & Jason M. Hall-Spencer [1,4]

Human activities are rapidly changing the structure and function of coastal marine ecosystems. Large-scale replacement of kelp forests and coral reefs with turf algal mats is resulting in homogenous habitats that have less ecological and human value. Ocean acidification has strong potential to substantially favour turf algae growth, which led us to examine the mechanisms that stabilise turf algal states. Here we show that ocean acidification promotes turf algae over corals and macroalgae, mediating new habitat conditions that create stabilising feedback loops (altered physicochemical environment and microbial community, and an inhibition of recruitment) capable of locking turf systems in place. Such feedbacks help explain why degraded coastal habitats persist after being initially pushed past the tipping point by global and local anthropogenic stressors. An understanding of the mechanisms that stabilise degraded coastal habitats can be incorporated into adaptive management to better protect the contribution of coastal systems to human wellbeing.

[1] Shimoda Marine Research Center, University of Tsukuba, 5-10-1 Shimoda, Shizuoka 415-0025, Japan. [2] Department of Botany, University of Otago, Dunedin, New Zealand. [3] The Marine Biological Association, Plymouth, Devon PL1 2PB, UK. [4] School of Biological and Marine Sciences, University of Plymouth, Plymouth PL4 8AA, UK. [5] These authors contributed equally: Ben P. Harvey, Ro Allen. ✉email: ben.harvey@shimoda.tsukuba.ac.jp

Human activities are causing widespread changes to the biota, chemistry and environmental conditions of the ocean[1]. While these changes are often gradual, they can reach tipping points, resulting in dramatic and abrupt regime shifts that change the structure and function of biological communities and are persistent over ecological time scales[2,3]. Regime shifts typically occur when a threshold (e.g. temperature, seawater pH, nutrient input or overfishing) is exceeded, triggering major changes in the internal dynamics of the ecosystem[2]. Internal dynamics are those processes within trophic levels (e.g. competition) or between trophic levels (e.g. resource-consumer interactions) that regulate and maintain ecosystem stability[4]. Regime shifts can cause large losses of ecological and economic resources, and yet due to their non-linear nature are often difficult to predict[5].

Regime shifts have been identified with increasing frequency across a range of marine ecosystems over the last few decades[6]. They are of particular concern since the newly established habitats are predominantly composed of species that possess less ecological, functional and human value compared to the habitats they replaced[7], and can persist for decades unless suitable management is put into place[8]. Coastal regime shifts have resulted in the large-scale replacement of spatially dominant foundation species (e.g. coral reefs, kelp forests and other habitat-forming macrophytes) with systems dominated by turf algae (e.g. refs. [9–11]). Turf algae tend to be fast-growing, opportunistic species with high turnover rates that tolerate stressful conditions well[12]. These multi-species assemblages of filamentous benthic algae can form dense mats or carpets across extensive areas that can trap large amounts of sediment[13]. Both global and local anthropogenic stressors can favour the growth and proliferation of turf algae, as well as negatively affect the growth and survival of canopy-forming algae and corals[14,15]. Finally, although turf algal mats can provide a highly complex habitat structure, it is on a fundamentally smaller scale compared to that provided by coral reefs and kelp forests[16] and so the shift to turf algae results in the loss of ecosystem services[17]. Here, we join the concerted effort to understand the processes responsible for driving and stabilising coastal regime shifts.

Among the internal mechanisms that shape an ecosystem, feedback loops play an important role in driving, maintaining and reversing regime shifts, due to their ability to both stabilise (negative feedback) and de-stabilise (positive feedback) ecosystem states[4,18]. Hysteresis describes the situation whereby once a system has crossed a tipping point and transitioned into a new state, the original state of the system cannot simply be recovered by an equivalent reversal of the environmental conditions; the environmental conditions instead need to be reversed beyond the initial tipping point[4,19]. It has been suggested that hysteresis explains the lack of recovery in both macroalgal-dominated coral reefs[20] and heavily grazed kelp forests[21,22]. Hysteresis has also been proposed as the reason why so few degraded turf states recover naturally[23]. Ecosystem managers try to avoid triggering regime shifts by assessing ecosystem resilience, as the restoration of those systems that are already degraded is exceedingly more difficult due to feedback loops[24,25]. The need to understand feedback loops is more relevant than ever as regime shifts become more common, highlighting an important need for research to identify their presence and relative strength.

As a global stressor, ocean acidification is projected to simplify coastal ecosystems as stress-intolerant species are lost[26–28]. This reshaping of biological communities typically means that biodiversity is reduced resulting in fewer ecosystem services[27,29]. The effects of ocean acidification are difficult to predict as it can act as both resource and stressor[29,30]. The increase in the availability of bicarbonate and $CO_2$ is a resource to primary producers[31] while the decrease in pH and calcium carbonate saturation state acts as a stressor to many other organisms via negative effects on their physiology and or shell production[32,33]. Volcanic $CO_2$ seeps offer opportunities to assess how marine communities are affected by long-term ocean acidification. Work at $CO_2$ seeps off Shikine Island, Japan[28,34] showed that seawater acidification caused coral and macroalgal habitats to be displaced by turf algae (Fig. 1). This simplified the ecosystem with a loss of diversity and habitat structural complexity (see Fig. 2 for a visual summary). Many $CO_2$ seep sites, across temperate, sub-tropical and tropical systems demonstrate similar coastal ecosystem simplification with increases in the abundance of macroalgae and turfs along gradients of increasing levels of $pCO_2$ in seawater[27–29,35–37].

Here, we support previous observations that $CO_2$ enrichment of coastal seawater favours turf algae over corals and macroalgae (see Supplementary Fig. 1 and Supplementary Table 1), and we examine whether this turf algal state is being held in place by stabilising feedback mechanisms and hysteresis. We expected that the presence of turf algae would alter the physicochemical environment within the turf algal mat. We therefore examine the spatial variability of the carbonate chemistry and dissolved oxygen, and the diurnal variability of pH. Changes in the environmental conditions can lead to (and be affected by) changes in the associated microbial community, and so we characterise the spatial variability in the microbial community composition within the turf algal mat by DNA metabarcoding using the 16S rRNA gene. Taken together, we predict that these changes in the physicochemical environment and microbial community may cause an inhibition in the recruitment of other macroalgal species and act as a feedback loop that reinforces the persistence of the

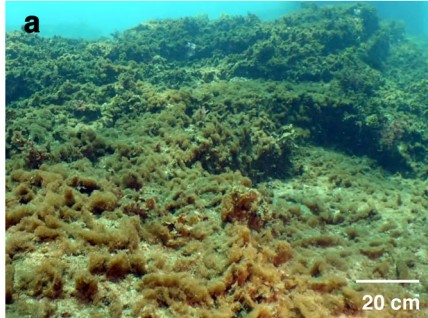
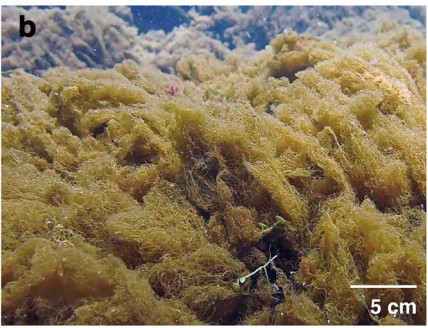

**Fig. 1 Extensive coverage of turf algae within the elevated $pCO_2$ areas of Shikine Island $CO_2$ seep. a** In elevated $pCO_2$ locations, turf algae covered most of the available space on shallow sublittoral rock. **b** By the end of summer, before the arrival of autumn typhoons, opportunistic fast-growing turf algae completely smothered rock surfaces. The predominant species forming the turf algal mat is *Biddulphia biddulphiana* (J.E.Smith) Boyer (for further description of the turf algal mat, see Supplementary Methods and Harvey et al.[34]).

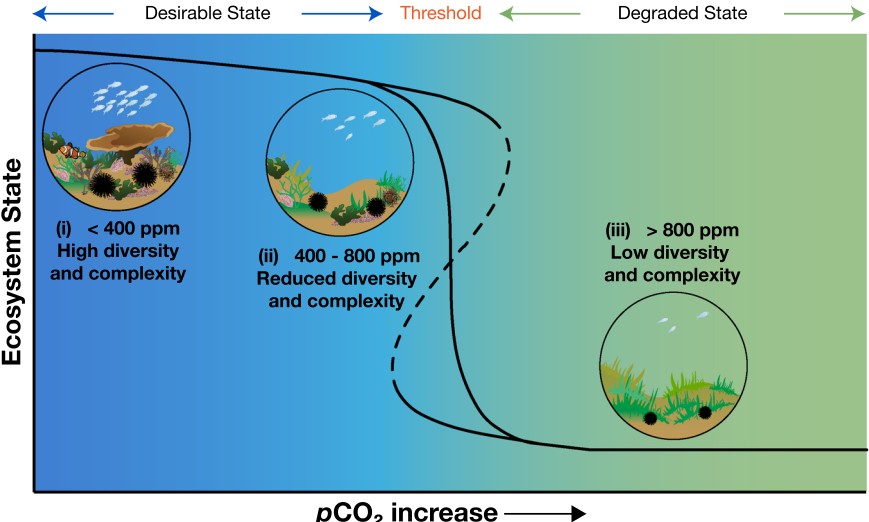

**Fig. 2 Ecosystem shifts with rising seawater $pCO_2$.** Ocean acidification drives changes that lower coastal biodiversity and habitat complexity. Black solid line highlights a continuous shift (Forward threshold = Reverse threshold) from the 'Desirable Ecosystem' with high diversity and complexity, to the 'Degraded Ecosystem' with low diversity and complexity. The black dashed line highlights a discontinuous shift (Forward threshold ≠ Reverse threshold) where feedback mechanisms can hinder ecosystem recovery ('hysteresis').

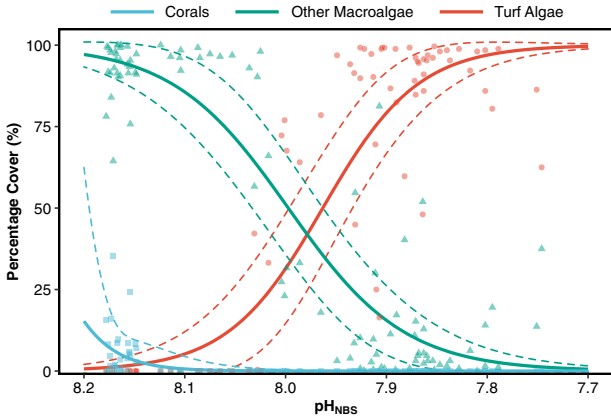

**Fig. 3 Influence of pH on the coverage of turf algae, other macroalgae and corals.** Mean cover (%) of turf algae (red circles), other macroalgae (green triangles) and corals (blue squares) along a seawater $pH_{NBS}$ gradient off Shikine Island, Japan. Solid lines represent a negative binomial generalised linear model (df = 99 biologically independent samples), with the dashed lines representing the 95% confidence interval. The pseudo-$R^2$ value of the three models are: turf algae 0.748, other macroalgae 0.773, and corals 0.541.

turf algae. The presence of feedback loops is a necessary condition for the emergence of hysteresis. By transplanting turf algal assemblages back along the $pCO_2$ gradient we assess for differences in the forward and reverse ($pCO_2$) threshold of turf algae in order to examine whether our system exhibits hysteresis.

## Results

**Change in dominant habitat type.** The abundance of turf algae in quadrats placed along belt transects was significantly greater at reduced pH (Fig. 3; GLM, $z = -5.27$, df = 98, $p < 0.001$; pseudo-$R^2 = 0.748$; Supplementary Table 2) changing from 0% cover in the reference $pCO_2$ sites to near 100% cover at elevated $pCO_2$ and vice versa for macroalgae (Fig. 3; GLM, $z = 5.84$, df = 98, $p < 0.001$; pseudo-$R^2 = 0.773$; Supplementary Table 2). Corals were sparsely distributed in the reference $pCO_2$ sites and completely

absent from all of the quadrats surveyed below $pH_{NBS}$ ~8.10. Subsequently, their abundance did not show a significant relationship with pH (Fig. 3; GLM, $z = 0.71$, df = 98, $p = 0.48$; pseudo-$R^2 = 0.541$; Supplementary Table 2). Overall, the change in abundance of each of the three groups demonstrated relatively abrupt (rather than gradual) changes over the pH gradient suggesting the presence of tipping points/thresholds (Fig. 3).

**Effects of simplified habitat on environmental conditions in the field.** Turf algae had a major effect on seawater chemistry within and beneath the turf algal mat, significantly reducing seawater pH (Fig. 4a; Kruskal–Wallis: $H_{3,56} = 33.5$, $p < 0.001$), dissolved oxygen (Fig. 4b; Kruskal–Wallis: $H_{3,56} = 48.3$, $p < 0.001$) and aragonite saturation state (Fig. 4c; Kruskal–Wallis: $H_{3,56} = 33.1$, $p < 0.001$) relative to the levels measured 2–3 m above the turf algal mat. There were no significant differences in seawater pH (post-hoc Dunn test, $p = 0.74$), dissolved oxygen (post-hoc Dunn test, $p = 0.14$) or aragonite saturation state (post-hoc Dunn test, $p = 0.79$) between the surface of the turf algae and 2–3 m above. Within the turf algal mat, pH was reduced by a mean of 0.43 $pH_{NBS}$ units, and both dissolved oxygen and aragonite saturation state were reduced almost twofold. In the sediment beneath the turf algal mat the mean $pH_{NBS}$ was $7.28 \pm 0.05$, dissolved oxygen levels approached hypoxia ($2.50 \pm 0.35$ mg l$^{-1}$) and there was aragonite undersaturation ($\Omega_{Aragonite}$ $0.56 \pm 0.06$). The average amount of sand being trapped by the turf algal mat was $969.1 \pm 180.4$ g m$^{-2}$ (S.E., $n = 6$ independent clearings of turf algae).

**Effects of simplified habitat on environmental conditions in laboratory microcosms.** Under light conditions, algal photosynthesis in the microcosms increased seawater pH from $pH_{NBS}$ $7.79 \pm 0.01$ up to a mean $pH_{NBS}$ of $8.17 \pm 0.06$ S.E. near the surface of the turf algal assemblage (Fig. 5a). The pH fell steeply to a mean value of $7.02 \pm 0.11$ S.E. within the turf and into the sediment (Fig. 5a). In the dark, pH was relatively constant within the turf at ~7.5 (Fig. 5a). During 3-h light-dark cycles, a change from a steady-state light pH to a steady-state dark pH and vice versa occurred over ~1.5 h after the light was changed, resulting in a mean change of ~0.8 $pH_{NBS}$ units in the water within the upper

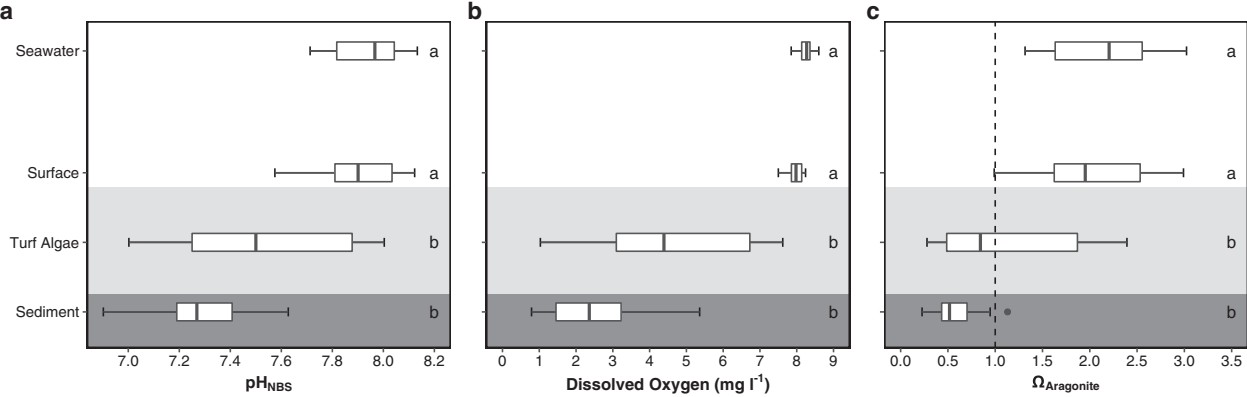

**Fig. 4 Spatial variability of the carbonate chemistry and dissolved oxygen associated with the turf algae.** Box plots ($n = 15$ biologically independent locations) of **a** $pH_{NBS}$, **b** dissolved oxygen (mg l$^{-1}$), and **c** aragonite saturation state of seawater ($\Omega$). Sediment = sediment below the turf algal mat (indicated by darkly shaded region), Turf Algae = middle of the turf algal mat (indicated by lightly shaded region), Surface = surface of the turf algal mat, Seawater = water column 2–3 m above the turf algal mat. Note that on **c** aragonite dissolution is favoured below $\Omega = 1$ (dashed line). For the boxplots, the line inside indicates the median, the upper and lower hinges correspond to the interquartile range, and the upper and lower whiskers extend to the highest and lowest values that are within 1.5 × the interquartile range of the hinge. Outliers beyond these whiskers are indicated with a point. A significant difference between the locations is indicated with a different letter (Kruskal–Wallis with post-hoc using a Dunn test) on the right of each panel.

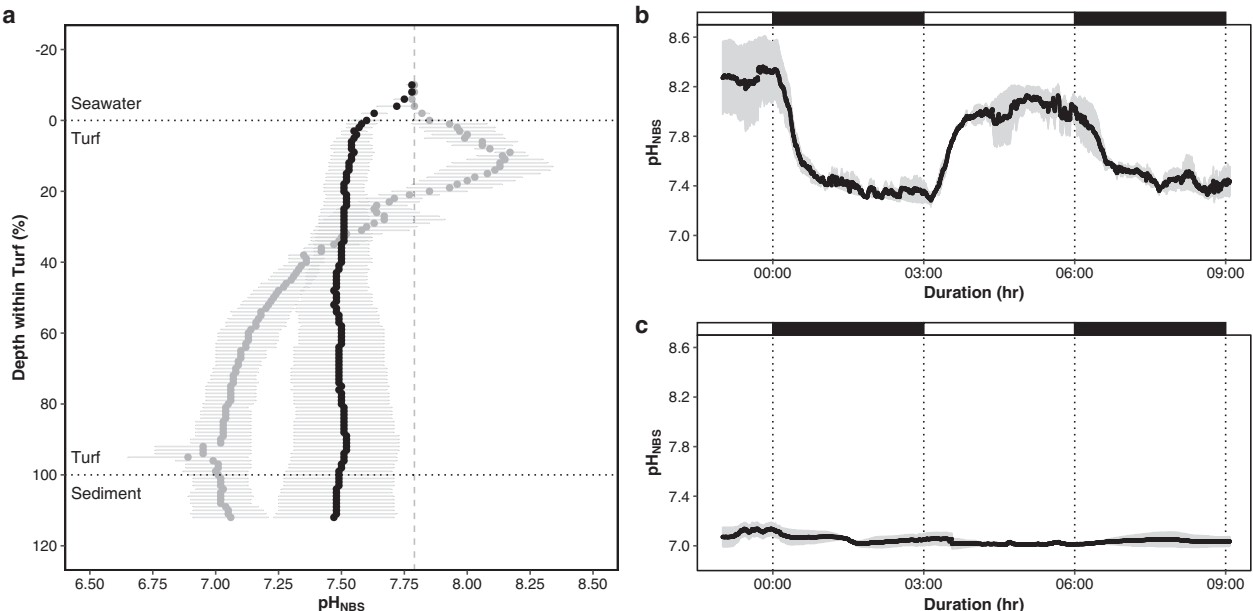

**Fig. 5 Microchemical environment of turf algae and associated sediments, and its diurnal variability. a** $pH_{NBS}$ profile (mean ± S.E., $n = 3$ independent microcosms) within the turf algae and sediment in the laboratory incubations. Grey circles show measurements made in the light period, black circles show measurements made in the dark period. Note: 0% refers to the surface of the turf algae. **b, c** $pH_{NBS}$ measurements (mean ± S.E., $n = 3$ independent microcosms) during 3-h light-dark cycles in the laboratory incubations (white shading above: light period, black shading above: dark period) at the point of maximal pH in the upper layer of the turf algae (**b**) and in the sediment below the turf algae (**c**).

layer of the turf assemblage, just below its surface (Fig. 5b). In the sediment below the turf algal assemblage the light-dark shift had minimal effect on pH, with $pH_{NBS}$ averaging 7.06 ± 0.04 (S.D.) across the light and dark periods (Fig. 5c).

**Effects of altered habitat type on associated microbial communities.** Distinct microbial communities were associated with the surface and middle of the turf algal mat, and the sediment below (PERMANOVA: pseudo-$F_{2,22} = 12.85$, $R^2 = 0.56$, $p < 0.001$; pairwise comparisons: surface-middle, $p = 0.003$; surface-sediment, $p < 0.001$; middle-sediment, $p < 0.001$; Fig. 6a). Microbial communities at the surface of the turf algal mat were dominated by heterotrophic Alphaproteobacteria, Bacteroidetes and Gammaproteobacteria. The relative abundance of Deltaproteobacteria increased in the middle

of the turf algal mat, before reaching dominance in the sediment below (Fig. 6b). ASVs belonging to the anaerobic sulfate-reducing Deltaproteobacteria, and sulfur-oxidising Epsilonbacteraeota, had the greatest relative abundance in the sediment below the turf algae (Supplementary Figs. 2 and 3).

**Effects of altered habitat type on recruitment of other macroalgae.** The presence of turf algae significantly reduced the recruitment of crustose coralline algae (CCA) onto tiles (Fig. 7a; ANOVA: $F_{2,7} = 267.3$, $p < 0.001$), despite all tiles being held at the same pH conditions. In the absence of the turf algae (Fig. 7b), CCA grew over 85% of the tile, whereas partial (Fig. 7c) and total (Fig. 7d) cover by the turf algae reduced CCA recruitment to ~45% and <1% cover, respectively (Fig. 7). A number of other

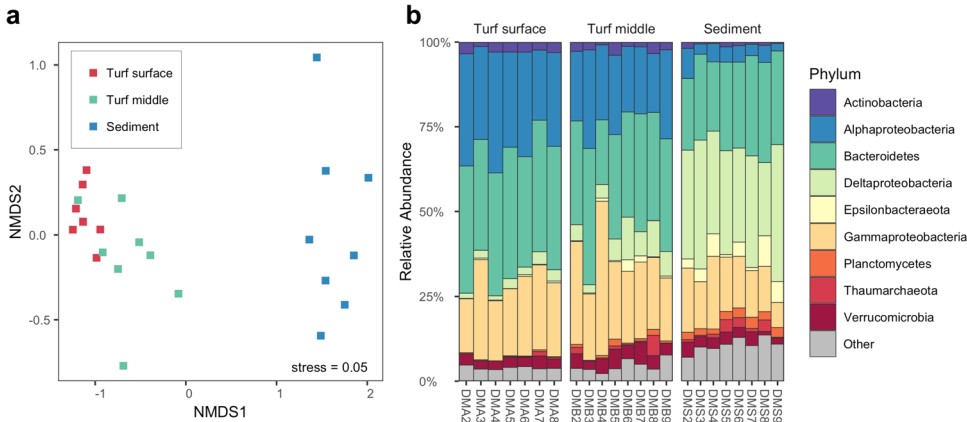

**Fig. 6 Microbial community structure associated with turf algae. a** nMDS plot of microbial community structure associated with turf algae based on Bray–Curtis dissimilarity ($n = 8$ biologically independent locations). Colours indicate the position within and below the turf algal mat from which samples were collected. **b** Stacked bar plot of microbial community composition at the phylum level and Proteobacterial class level. The nine most abundant phyla based on total number of reads are displayed, additional phyla are grouped as 'Other'.

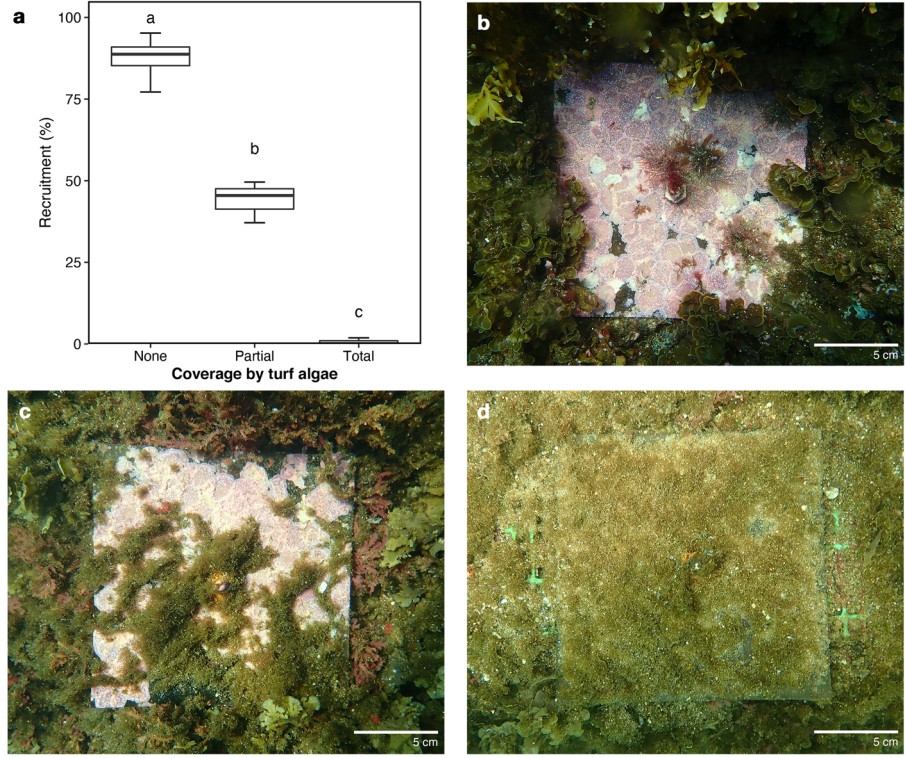

**Fig. 7 Influence of turf algae on the recruitment of other algae. a** Boxplots showing the effects of the presence of turf algae on the recruitment of crustose coralline algae. The presence of the turf algae was classified as no coverage 'None' (see **b**, $n = 4$ biologically independent tiles), partial coverage 'Partial' (see **c**, $n = 3$ biologically independent tiles) or total-coverage 'Total' (see **d**, $n = 4$ biologically independent tiles). For the boxplots, the line inside indicates the median, the upper and lower hinges correspond to the interquartile range, and the upper and lower whiskers extend to the highest and lowest values that are within 1.5 × the interquartile range of the hinge. A significant difference between the coverage groups is indicated with a different letter (ANOVA with Tukey's HSD post-hoc test).

small macroalgae recruits were also observed on the recruitment tiles in the absence of the turf algae (<5% cover, Fig. 7b), but were not found on any of the tiles that were partially or fully covered by turf algae (Fig. 7c, d).

**Testing for differences in the forward and reverse threshold along the $p$CO$_2$ gradient.** At the reference $p$CO$_2$ location (pH$_{NBS}$ 8.17 ± 0.001 S.E.), where the turf algae are very low in abundance, the transplanted turf algal assemblage no longer persisted after 1-month (Fig. 8). The percent cover of the transplanted turf algal assemblage at the least CO$_2$-enriched location (pH$_{NBS}$ 8.04 ± 0.04) persisted after 2-months (83.1% ± 5.6) despite natural levels of turf algae in the area being low (4.2% ± 4.2; Fig. 8). At the other elevated $p$CO$_2$ locations (pH$_{NBS}$ 7.91 ± 0.04, 7.87 ± 0.02, 7.85 ± 0.02), natural levels of turf algae were considerably higher (74.7% ± 9.7, 94.1% ± 2.2 and 98.8% ± 0.2 respectively) and the percent cover of the transplanted turf algal assemblage remained above 95% after 2-months (Fig. 8).

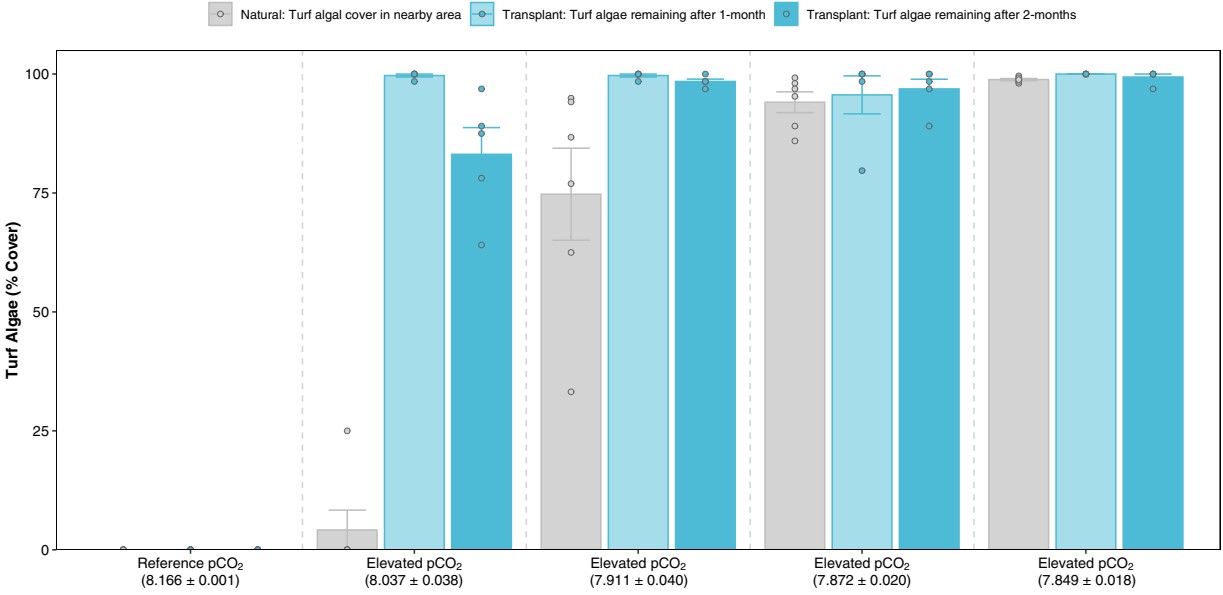

**Fig. 8 Reverse threshold of turf algae transplanted back along the pH gradient compared to their natural percentage cover.** Natural turf algal cover (grey) and turf algal assemblage remaining following a 1-month (light blue) and 2-month (dark blue) transplant of pre-established (fully covered) turf algal assemblage tiles into each site (indicated by their mean pH$_{NBS}$ ± S.E.). Natural turf algal cover ($n = 6$ biologically independent measurements at each time point, summarised from Fig. 3) and the turf algal assemblage transplants ($n = 5$ biologically independent tiles at each site) are shown as mean ± S.E.

## Discussion

Ocean acidification can alter the dominance of habitat-forming species[38] acting both as a stressor to, and a resource for, different members of the biological community[29,30]. Turf-forming algae are suppressed by grazing on healthy coral reefs[20] and underneath macroalgal canopies on temperate reefs[39]. When either released from competition with other habitat-forming species[20,40], or when their growth is exceeding the capacity of grazers to control (e.g. Refs. [29,41]), turf algae can proliferate and become dominant. Corals are particularly sensitive to the direct effects of ocean acidification[42,43], and can be outcompeted by macroalgae under acidified conditions[44]. For most macroalgae, the additional carbon can act as a resource promoting their growth[31]. However, not all primary producers will respond to this resource enhancement evenly[30]. Here, we found that as seawater acidification caused the decline of corals and large macroalgae, there was a major increase in turf algae abundance, indicating that they are capable of monopolising the available space and becoming the most abundant habitat-forming species, corroborating previous works[28,34]. These changes in the abundance of the corals, macroalgae and turf algae along the $p$CO$_2$ gradient were abrupt rather than gradual shifts, suggesting that the environmental conditions exceeded a critical threshold[4]. Taken together, such anthropogenic-driven shifts are causing widespread loss in coastal habitat structural complexity as coral reefs and kelp forests are shifting to systems dominated by turf algae.

Once established, turf algae can physically inhibit the availability of suitable substrata (e.g. coralline algae covered rock) for the settlement of macroalgae[45,46] and corals[47,48]. In the present study, turf algae trapped large quantities of sediment which then often became anoxic. This turf-trapped sediment can act as a physicochemical barrier for settlement[49] and impair the growth and photosynthetic activity of other organisms[50]. The trapping and binding of sediments is a common process in mats of turf algae[12,50], where excreted exopolymers help bind and stabilise the sediment[51]. As well as physically impacting settlement success, sedimentation can also affect the success of other organisms by causing abrasion, scouring and burial, potentially damaging or removing organisms[52]. Sedimentation can also physically reduce the light[53], causing changes in the characteristics of the chemical microenvironment such as creating anoxic conditions[50]. Here we observed reduced levels of pH, oxygen and aragonite saturation state both within the algal turf mat and the sediment underneath. These mechanisms could result in the inhibition of recruitment either by reducing settlement viability or by increasing post-settlement mortality. Given the observed reduction in the abundance of both macroalgae and coral species within the elevated $p$CO$_2$ area, it is likely that the potential inhibition of recruitment may be further limited by low propagule supply due to the reduced abundance of adults in the nearby area. Both mechanisms would, however, contribute to the hysteresis maintaining the observed regime shift into a turf-dominated state.

The activity of microbial communities associated with the turf algae is likely to be a driver of the altered chemical microenvironment observed in this study. Microbial communities at the surface and middle of the algal turf mat were dominated by aerobic heterotrophs (Fig. 6b and Supplementary Fig. 2), likely fuelled by a supply of photosynthetic exudates that will have been released by the turf algae[54,55]. Microbial respiration likely underpins rapid decreases in oxygen concentration and pH, creating hypoxic conditions below the turf algae which is known to be deleterious to habitat-forming organisms such as scleractinian corals[56]. The hypoxic sediments below the turf algal mat had a microbial community dominated by sulfate-reducing Deltaproteobacteria and sulfur-oxidising Epsilonbacteraeota. The activity of these sediment-associated anaerobic microbes likely contributes to further pH decreases, as well as the accumulation of hydrogen sulfide (H$_2$S) indicated by the characteristic black colouration and smell of these sediments (Supplementary Fig. 4) which can inhibit the recruitment of canopy-forming macroalgae[53]. Taken together, the activity of microbes associated with turf algae is likely to contribute significantly to the establishment and maintenance of the altered chemical microenvironment. Indeed, photosynthetic exudates released by turf algae have previously been shown to support higher microbial biomass[57,58], thus enhancing microbial-driven alterations to chemical

microenvironments (i.e. hypoxia). Increased amounts of algal-derived dissolved organic content (DOC) can also bring about a decline in corals by promoting opportunistic pathogens through a positive feedback loop termed the DDAM model (Dissolved organic matter (DOM), disease, algae and microbes) (reviewed in Ref. [59]). Overall, these conditions inhibit corals and canopy-forming algae, creating a feedback mechanism by which the competitive superiority of turf algae is reinforced through microbial vectors[57].

The novel conditions created by the turf algal mat are likely to alter which species will be facilitated and may result in changes to the trophic structure. Previous work has demonstrated that shifts to turf algal mat can enhance the diversity of small invertebrates being facilitated when compared to habitat provided by larger macroalgae species[60]. Here we found that the conditions within the turf algae varied diurnally, showing high primary productivity near the turf surface during light conditions (increasing ~+0.4 $pH_{NBS}$ units compared to the seawater) but reaching a minimum during the dark conditions (decreasing ~0.3 $pH_{NBS}$ units relative to the seawater). Within the sediment, the variations in pH over 3-h light/dark shift periods were minimal compared to the highly productive surface of the turf assemblage (Fig. 5b, c), although it might exhibit greater changes over diurnal periods. These changes to the chemical microenvironment mean that those organisms residing within the turf algal mat will experience spatially variable pH, as evidenced by the gradients in pH, DO, aragonite saturation from the turf algal mat surface to the sediment below, as well as daily pH fluctuations that both far exceed the long-term trend predicted for the open ocean. The most abundant taxon of the mobile invertebrates in the turf algal mat were tanaid crustaceans (~50% of total abundance in the high-$CO_2$ and ~10% in the reference $CO_2$, Harvey et al.[34]), which can often be found in anoxic conditions[61], as well as at other naturally acidified systems (Papua New Guinea $CO_2$ seep; Allen et al.[62]). This lower species evenness suggests that only the stress-tolerant taxa may be facilitated under elevated $pCO_2$ conditions[26]. As turf algae become the most abundant habitat, the change to the micro-environment will have a major impact on habitat structure as well as trophic food webs associated with these species.

Human activities can mediate turf transitions through a number of different abiotic (ocean acidification, warming, eutrophication) and biotic (herbivory and epiphytism) drivers[15,23]. Although the initial external impact responsible for transitioning ecosystems to a turf-dominated state will differ for each of the drivers, the stabilising feedback mechanisms responsible for locking the turf algae transition in place show similarities across these abiotic and biotic drivers[16,23,50]. The transplant of turf algal assemblages from the elevated $pCO_2$ area back along the $pCO_2$ gradient highlighted that the forward threshold (as shown by the abrupt increase in the natural abundance of turf algae as $pCO_2$ increases) differed from the reverse threshold (as shown by the persistence of the transplanted turf algal assemblages under moderate $CO_2$ enrichment where turf algal mats do not naturally develop) of our ecosystem state (Fig. 8). The difference between the forward and reverse threshold, combined with the stabilising mechanisms maintaining the turf-dominated state here, suggests that this ecological shift may exhibit hysteresis, and would therefore be particularly difficult to reverse. Ultimately, although the initial regime shift here was driven by ocean acidification, the observed mechanisms will be equally relevant in other scenarios.

For ecosystem management, two overarching strategies exist when considering ecosystem shifts; reversing an ecological shift by returning the environmental conditions to prior conditions, or mitigating the future degradation of ecosystems. The issue with attempting to reverse an ecological shift is that whereas a localised impact (such as eutrophication) is reversible through appropriate management, future climate change projections (e.g. IPCC[63]) would suggest that for global impacts (such as ocean acidification) reversal on timescales relevant to humans is unlikely[23]. Our case study here represents a window into a potential future ecosystem state where an ecosystem has already greatly exceeded its critical threshold. The observed stabilising feedbacks that lock the turf system in place would likely necessitate more active measures to manipulate turf algal mats and facilitate recovery once a regime shift has occurred. While feasible, such an approach is unlikely to present a particularly scalable solution. We suggest that it is more tangible to mitigate future degradation by promoting the resilience of intact desirable ecosystems. One such approach of enhancing the resilience of coral/macroalgal habitats is to ameliorate locally-manageable human impacts (e.g. fisheries, eutrophication) in order to prevent harmful synergisms with global stressors like acidification that are less manageable at a local level[64]. Ecosystem resilience building may also be achievable through augmentative biocontrol, for example, with sea urchins previously trialled as biocontrol agents of turf algae on coral reefs[65,66], or via the manual removal of undesirable macroalgae[67] (the spatial scalability of such solutions is still being assessed). Understanding when management actions are required will be key; foreseeing ecological shifts will require the identification of early warning indicators that signal the approach of an abrupt tipping point. Continued degradation of ecosystems highlights a need for proactive management, which must incorporate the dynamics of ecosystems in order to mitigate against future degradation.

In conclusion, our findings show that seawater acidification changes the relative abundance of key functional groups in marine ecosystems with turf algae being favoured, and that reinforcing feedback loops play an important role in maintaining systems dominated by turf algae. Many $CO_2$ seep sites across temperate, sub-tropical and tropical systems have also demonstrated increases in the abundance of turf algae under elevated levels of $CO_2$[28,29,37]. Similar feedback loops (and resulting hysteresis) are likely to be responsible for similar shifts to turf algae across biogeographic regions and stressors. The ability of management to mitigate or reverse an ecological regime shift depends upon an understanding of how species interactions stabilise marine ecosystems. Regime shifts are projected to increase in frequency and distribution, due to both local and global drivers, and our work contributes towards enabling better science-based management of regime shifts.

## Methods

**Study site.** We assessed the responses of shallow (0–10 m depth) coral and algal habitats along a gradient of seawater $CO_2$ concentrations at $CO_2$ seeps off Shikine Island, Japan (34°19′9″ N, 139° 12′18″ E). The study area comprised rocky reef habitat with a reference $pCO_2$ location characterised by a mixture of both canopyforming fleshy macroalgae and zooxanthellate scleractinian corals. Elevated $pCO_2$ locations (under the influence of the $CO_2$ seep) were characterised by turf algal mats (Fig. 2). A description of all of the sites along with their carbonate chemistry is provided by Agostini et al.[28] and details about the turf algae are provided by Harvey et al.[34]; this information is summarised in the Supplementary Methods. Our elevated $pCO_2$ areas represent end-of-the-century projections for reductions in pH (the RCP 8.5 scenario, IPCC[63]), and were not confounded by differences in temperature, dissolved oxygen, total alkalinity, nutrients or depth relative to reference sites used for comparison[28,34].

**Change in dominant habitat type.** Percentage cover of the substratum and associated carbonate chemistry were assessed in June 2017 using nine 50 m transects (three in the reference area and a further six following gradients in $pCO_2$), with four photoquadrats (50 × 50 cm) and a seawater sample (100 ml) taken every 5 m along the transects (transect locations shown on Supplementary Fig. 5). Seawater sample temperature, salinity and $pH_{NBS}$ were immediately recorded (Orion 4-Star, Thermo Scientific, USA) onboard RV *Tsukuba II* (University of Tsukuba research vessel). Photoquadrats were analysed using ImageJ[68] by overlaying 64

points on a grid, and recording the presence of turf algae, other macroalgae, or coral. The relationship between the percentage cover and measured pH was assessed using a negative binomial generalised linear model (Family: Binomial, Link: Logit) with the 'predict.glm' function in R[69].

**Effects of habitat simplification on environmental conditions in the field**. To assess indirect effects of turf algae on water chemistry, 15 sets of measurements were taken from haphazard locations where turf algae abundantly grows under the elevated $pCO_2$ conditions. For each set of measurements, four water samples were collected: (i) in the sediment below the turf algae, (ii) in the middle of the turf algal mat, (iii) at the surface of the turf algal mat, and (iv) in the seawater 2–3 m above the turf algae; hereafter referred to as 'Sediment', 'Turf Algae', 'Surface', and 'Seawater'. Water samples of 10 ml were collected using a length of tubing connected to a syringe, to allow samples to be taken by SCUBA divers with minimal disturbance. Samples taken with the syringe were collected slowly and carefully to avoid de-gassing. Water samples were then measured for temperature, $pH_{NBS}$, dissolved oxygen and salinity (Orion 4-Star, Thermo Scientific, USA) onboard RV *Tsukuba II* (University of Tsukuba research vessel). Following these measurements, the water samples were then filtered at 0.45 μm using disposable cellulose acetate filters (Dismic, Advantech, Japan) and stored at room temperature in the dark until total alkalinity ($A_T$) measurement. Total alkalinity was measured by titration (916 Ti-Touch, Metrohm) with HCl at 0.1 mol l$^{-1}$, and then calculated from the Gran function between pH 4.2 and 3.0. Carbonate chemistry parameters were calculated using the CO2SYS software[70]. Measured $pH_{NBS}$, $A_T$, temperature and salinity were used as the input variables, alongside the disassociation constants from Mehrbach et al.[71], as adjusted by Dickson and Millero[72], HSO$_4$ using Dickson[73], and total borate concentrations from Uppström[74]. The effects of the turf algae on seawater chemistry ($pH_{NBS}$, dissolved oxygen and aragonite saturation state) between 'Sediment', 'Turf Algae', 'Surface' and 'Seawater' were assessed using non-parametric Kruskal–Wallis with the Dunn test used for post-hoc comparisons in R[69], as data did not conform to the assumptions of normality and equality of variance.

The mean amount of sediment being trapped by the turf algal mat in-situ was assessed by carefully scrapping and collecting everything within a circular area (1 m diameter) into ziplock bags ($n = 6$) within the elevated $pCO_2$ area. The turf algae and associated sediment were separated by rinsing them through a 5.6-mm sieve (to catch the turf algae), with a 300-μm sieve placed underneath to catch the sediment (>300 μm). This sediment was dried for 72 h at 60 °C, and weighed (SJ 6200, Shinko Denshi, Japan). It should be noted that our approach is meant to be indicative and the estimated sediment weight within the turf algal mat will be slightly underestimated as some of the finer-grained sand (300–63 μm), silt (63–2 μm) and clay (<2 μm) could not reasonably be collected.

**Effects of simplified habitat on environmental conditions in laboratory microcosms**. Turf algal microcosms were set up in the laboratory to measure within assemblage pH profiles using microsensors for comparison with field observations. In the field, we noticed that by the time the summer season arrived turf algal mats had trapped around a 5-cm-thick layer of sediment on rock surfaces. We simulated turf algal assemblages ($n = 3$ independent microcosms) by placing a 5-cm layer of sediment collected from underneath turf algae at our study site into a vertical (30 cm $H \times 10$ cm $D$) clear Perspex tube and overlaying this with a 25-mm-thick layer of turf algae (also from the study site). Black vinyl tape was placed around the sediment layer and lower half of the turf algae as only light penetration from above was possible in the field. Each turf algal microcosm was continuously supplied with acidified seawater at $pH_{NBS}$ 7.779 ± 0.005 (S.D., $n = 3$ turf algal microcosms, with measurements taken daily) by bubbling $CO_2$ controlled using a Fukurow pH Controller (Aqua Geek, Kawaguchi, Japan). The microcosms were left to settle for 1 week before microsensor measurements were undertaken.

Our pH microsensor (Unisense pH-50 microsensor, Aarhus, Denmark) was mounted on a manually operated micromanipulator (MMN-333, Narishige, Tokyo, Japan). Measurements were carried out in the following order. We took a pH profile under light conditions (MT-250W Metal halide lamp, MzOne Co., Osaka, Japan) at 200 μm increments from 25 mm above the mesocosm turf algae into the simulated turf assemblage. Upon reaching the sediment, the microsensor was left in place for 1-h, and then the pH within the sediment below the turf algae was measured using 3-h alternating light-dark cycles (with measurements taken every 10 s) for 9-h. Following this, the point of maximal pH (near the surface of the turf algae) was located, and after 1-h, pH was measured for another set of 3-h alternating light-dark cycles (with measurements taken every 10 s) over 9-h. These 3-h periods for the light-dark cycles were sufficient to reach a stable pH value. Once complete, the turf algal microcosm was dark-adapted for 3 h, then a second pH profile was performed under dark conditions in a different part of the turf algal microcosm (at 200 μm increments from 25 mm above the mesocosm turf algae into the simulated turf assemblage). All measurements were performed on the same microcosm, and replicated three times using separate turf algal microcosms. Before every pH profile the microsensor was calibrated to the NBS pH scale using three buffers 4.01, 7.00 and 10.01 (Thermo Scientific, USA). Water flow was stopped during the pH profiles and light-dark cycles, meaning that measurements were carried out under static water conditions. Variation in the pH of overlying seawater was assessed by taking a measurement using the microsensor and a pH meter

(Orion 4-Star, Thermo Scientific, USA). Variation in pH of overlying water (±0.01 $pH_{NBS}$ units) was minimal during our measurements.

**Effects of altered habitat on associated microbial communities**. To characterise the composition of microbial communities associated with turf algal mats, samples were collected by hand from: (i) turf algae located in the upper part of the turf algal mat, (ii) turf algae from within the turf algal mat, and (iii) the sediment below the turf algal mat, from eight haphazard locations within an elevated $pCO_2$ area (24 samples in total). Turf algae or sediment was collected in 2 ml cryovials and then frozen in liquid nitrogen until DNA extraction. DNA was extracted using a Qiagen DNeasy Plant Mini Kit (Qiagen, Valencia, CA, USA) according to the manufacturer's instructions. The 515 F (5′-GTGYCAGCMGCCGCGGTAA-3′)[75] and 806R (5′-GGACTACNVGGGTWTCTAAT-3′)[76] universal primers, with a complementary region added to the 5′ end to facilitate the attachment of Illumina adaptors and indexes[77], were used to amplify the V4 region of the 16S rRNA gene. For first round PCR, triplicate 25 μl reaction mixtures were prepared for each sample using the HiFi Hotstart PCR kit (KAPA Biosystems, Boston, MA, USA). The first round PCR protocol had an initial denaturation step at 95 °C for 120 s, followed by 25 cycles of 98 °C for 20 s, 50 °C for 30 s and 72 °C for 30 s, with a final extension step at 72 °C for 60 s. First round PCR products were pooled and purified using the Mag-bind TotalPure NGS kit (OMEGA Bio-tek, Norcross, GA, USA). A second round of PCR attached the Illumina adaptors and indexes as described by Griffith et al.[77]. Second round PCR products were purified, pooled, and submitted for sequencing on the Illumina MiSeq platform using the V2 reagent kit (Illumina, San Diego, CA, USA).

Raw sequences were processed using R[69] according to the Bioconductor workflow for microbiome analysis[78], utilising the DADA2 algorithm to resolve amplicon sequence variants (ASVs) at single-nucleotide resolution[79]. Taxonomy was assigned to each sequence against the SILVA 132 database[80] using the RDP naïve Bayesian classifier[81]. Processed sequences were assembled as a phyloseq object[82] and randomly subsampled to an even depth of 13,807 reads prior to downstream analysis.

Differences in microbial community composition between the surface of the turf algal mat, the middle of the turf algal mat, and the sediment below the turf algal mat, were tested using a permutational analysis of variance (PERMANOVA) based on Bray–Curtis dissimilarity, implemented through the *adonis* function in the R package 'vegan'[83].

**Effects of altered habitat type on recruitment of other macroalgae**. We attached ten PVC tiles (150 × 150 mm) to sublittoral rock at 6–7 m depth in an elevated $pCO_2$ area at the end of March 2018 (at the beginning of the spring growth of turf algae). Tiles were separated by at least 3 m horizontal distance and were placed into the same light and wave exposure conditions. The tiles were retrieved in July 2018 and the turf gently removed to photograph any recruits underneath (Nikon D7200, Nikon, Japan). The percentage cover of new recruits of crustose coralline algae was assessed using ImageJ, comparing tiles that had been fully covered by turf algae (94.78% ± 1.96 S.D., $n = 4$ biologically independent tiles), partially covered by turf (46.01% ± 7.13 S.D., $n = 3$ biologically independent tiles), and those with no turf (0.00% ± 0.00 S.D., $n = 3$ biologically independent tiles). The difference in turf algal coverage between these three treatments (fully covered, partially covered, and no cover) significantly differed (Kruskal–Wallis, $H_{2,10} = 8.54$, $p = 0.01$). The effects of the turf algae on recruitment were assessed using ANOVA, with data conforming to the assumptions of normality (QQ plot) and equality of variance (Bartlett's) following a square root transformation. Post-hoc testing was assessed using Tukey's HSD.

**Testing for differences in the forward and reverse threshold along the $pCO_2$ gradient**. We attached plastic green mesh tiles (150 × 150 mm) to the sublittoral rock at 6–7 m depth in the most elevated $pCO_2$ area (based on the transects used in the 'Change in dominant habitat type' subsection) in March 2019 (at the beginning of the spring growth of turf algae). Tiles were separated by at least 3 m horizontal distance and were placed into the same light and wave exposure conditions. Once the tiles were fully covered by turf algal assemblages in late May 2019, the tiles were then transplanted back along the $pCO_2$ gradient. The transplanted tiles were attached to the sublittoral rock at 6–7 m at five different locations (Reference $pCO_2$, $pH_{NBS}$ 8.17 ± 0.001; and four elevated $pCO_2$ locations, $pH_{NBS}$ 8.04 ± 0.04, 7.91 ± 0.04, 7.87 ± 0.02, 7.85 ± 0.02 respectively), with five replicates at each location (25 tiles in total). The percentage cover of the remaining turf algal assemblage was assessed after 1-month (late June) and 2-months (late July) taking a photograph of each tile in-situ (Olympus TG-5 Camera). Analysis was performed using ImageJ by overlaying 64 points on a grid, and recording the presence of turf algae (using the same approach used in the section 'Change in dominant habitat type').

**Statistics and reproducibility**. All statistical analyses were carried out in R[69], using the base R, 'rstatix'[84], 'vegan'[83], and 'phyloseq'[82] packages for statistics, and the 'tidyverse'[85] and 'ggpubr'[86] packages for figure production. The specific statistical analyses employed are listed in their respective sections above. We have assessed the dynamics of the turf algae in the volcanic seep off Shikine Island

seasonally through scuba diving from 2015 to 2020, observing the same patterns every year.

**Reporting summary**. Further information on research design is available in the Nature Research Reporting Summary linked to this article.

## Data availability

Raw data used for Figs. 3–8 are freely accessible and stored on figshare[87]. Raw DNA sequences used for microbial community analyses are accessible through the European Nucleotide Archive (accession: PRJEB40334).

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

## Acknowledgements

We thank the technical staff at the 'Shimoda Marine Research Center, University of Tsukuba' for their assistance. This project contributes towards the 'International Educational and Research Laboratory Program', University of Tsukuba and made extensive use of RV *Tsukuba II*. This work was partially supported by Japan Society for the Promotion of Science (JSPS) KAKENHI Grant Number 17K17622, and we acknowledge funding from the Ministry of the Environment, Government of Japan (Suishinhi: 4RF-1701).

## Author contributions

B.P.H. conceived the idea, performed the image and statistical analysis, deployment and retrieval of recruitment tiles and drafted the manuscript. B.P.H., R.A., S.W., K.K. and J.M.H.-S. conducted the field surveys and sampling. B.P.H. and S.A. performed the microsensor analyses. R.A., L.J.H. and T.C.S. performed the molecular analysis of the microbial communities. All authors helped write the manuscript.

## Competing interests

Linn J. Hoffmann is an Editorial Board Member for Communications Biology, but was not involved in the editorial review of, nor the decision to publish this article. The authors declare no additional competing interests.
