## [Peer Review File · Communications Biology]

Reviewers' comments:

Reviewer #1 (Remarks to the Author):

The manuscript 'Feedback mechanisms lock degraded turf algal systems in place' by Harvey et al. focuses on how coastal ecosystem (coral reefs are hereby used as the main example) are subjected to feedback loops during ecosystem degradation.

The study sums up most of the existing work on feedback mechanisms (I think Barott and Rohwers DDAM model is missing) and presents the main points in a nice and concise form. The authors further corroborate the information drawn from the literature with their own findings from a study conducted at CO₂ seeps at Shikine Island, Japan.

All together this is a solid study that covers the existing literature and the experiments are straight forward. There are some minor points I would like to have addressed (see below), and one main issue:

These 2 sentences are in my opinion the core of the work: "The stabilising mechanisms maintaining the turf-dominated state here suggests that this ecological shift may exhibit hysteresis, and be more difficult to reverse. Management will need to either promote the resistance and resilience of the existing habitats (e.g. reducing local human pressure), or manipulate turf assemblages to promote recovery of the desirable state (i.e. coral and macroalgae dominated) following the transition"

My personal opinion here is that this is not necessarily groundbreaking new information. What I would like to see is some sort of (even if it is only speculative) more tangible advice or suggestions on how to improve protection or restauration strategies. The authors have shown that the feedback loops play a significant role – what are the implications of this information? What would be some hypothetical actions (based on their findings) that could break these feedback loops? If you would have to restore a degraded reef, how would you go about it, knowing about these feedback loops? I know this is not easy to include, because you have not done the restoration part and it is all just hypothetical. But (my opinion, the editors might have a very different view) if this work should be published in a journal of this quality I think it is important to provide a thought-provoking impulse to make it interesting for a wider audience.

Some minor points:

I think the image the authors use in figure 1b depicts the brown algae hydroclathrus. I would not necessarily call it a turf algae. (Turf algae (or "algal turfs") are dense, multi-species assemblages of filamentous benthic algae, including small individuals of macroalgae and cyanobacteria.)

Methods, microbial sampling. Samples were collected from the middle and the surface of the turf algae. This is somehow vague, I assume a tissue sample was taken from the upper part of a turf algae mat or consortium and one from within one of the turf algal mats, correct? That would make sense, but it is not stated clearly because now I read it as if it was taken from the algal surface and within an alga. But I know that is not what is meant...

Over all I think this manuscript deserves publication, but I would urge the authors to work on the discussion a bit as described above and be a bit more innovative with their conclusions and recommendations

Reviewer #2 (Remarks to the Author):

In this study, Harvey and colleagues provide a detailed and mechanistic insight into the processes that could stabilize regime shifts (large-scale and persistent ecosystem reorganizations) in a subtropical marine ecosystem. They find a continuous change in habitat structure along a gradient in pCO₂ at a natural CO₂ seep site in which hard corals and larger macroalgae that enhance 3-dimensional habitat structure are replaced by expansive mats of low-lying turf-forming algae. This provides a window into the type of simplified communities that can be expected to persist with ocean acidification under a 'business-as-usual' climate scenario. From field sampling and laboratory microcosms, they find that seawater carbonate chemistry and dissolved oxygen are altered within the layer of turf algae and accumulated sediments compared to the algal surface and overlying seawater. They also document a gradient in microbial community structure from the surface of the turf algae, to the middle, to the sediment below, which may be responsible for changes to the chemical microenvironment. These microbial-induced chemical changes may make conditions unsuitable for the settlement and survival of other sessile organisms and form an important feedback mechanism that locks the turf ecosystem state in place. This study also found that settlement of crustose coralline algae on PVC tiles, a favourable substrate for settlement of corals and macroalgae, was reduced under dense covers of turf algae.

The study is overall well done and nicely balanced, combining evidence from multiple sources (field sampling, laboratory, water chemistry, genetics) to support a well-rounded and cohesive narrative. The field sampling and laboratory components were appropriately designed and executed. Figures are neatly presented and attractive. Analyses were done properly. The quality of the writing is generally good and clear, but requires more focus and more nuanced interpretation in areas. My main criticism is that I don't think the authors have framed the paper to highlight the novel aspects of the study as well as they could. Other studies have already documented decreased abundance of foundation species like corals and enhanced productivity of turf algae along pCO₂ gradients at seep sites (e.g. Fabricius et al. 2011, Connell et al. 2013). Previous studies have also shown indirect effects of turfs on seawater chemistry (e.g. dissolved oxygen) induced by microbial respiration (Smith et al. 2006, Barott et al. 2009, Wangpraseurt et al. 2012, Jorissen et al. 2016) and shifts in the microbial community at coral-turf interaction zones (Barott et al. 2012). Furthermore, the negative effects of turf algae on recruitment of foundation species are well documented (e.g. corals: Birrell et al. 2005, kelp: Connell & Russell 2010, see O'Brien & Scheibling 2018 for meta-analysis of competition experiments). However, this study is the first to my knowledge to document effects of turf algae on pH and aragonite saturation state and improves on these previous studies by identifying a gradient in carbonate chemistry and microbial communities from the surface of the turf down through the layer of bound sediments. I'm just not sure it's enough to warrant publication in this forum. A general ecological journal may be better suited. I applaud Harvey and co-authors for the amount and quality of work they have accomplished. I think this will make an important contribution to the fields of ecology and conservation. It will be of wide interest to researchers and managers working in diverse marine systems from temperate and tropical regions alike where persistent replacement of foundation species by turf communities (and associated loss of ecosystem services) are becoming increasingly more commonplace. I would be happy to read a revised version of this manuscript whether resubmitted here or another journal. Specific comments are provided below.

1. L1: Title is a bit too general in my view. Consider revising to convey the main outcome(s) of the study and better inform the reader.
2. L28: 'increase turf algae' is a bit vague. Perhaps 'favour turf algae growth' or 'increase turf algae abundance'?
3. L31: The results of this study do not explicitly demonstrate competition for light and space and

increased sediment load. Please revise.

4. L32-34: Consider revising. Feedback loops can maintain, stabilize, or accelerate degraded shifts to ecosystem states, but often are disconnected from the local to global scale anthropogenic stressors that initially pushed the ecosystem passed a tipping point. Perhaps focus more on the persistence aspect.

5. L52: Should be 'habitats are predominantly composed of species' or 'habitats predominantly comprise'

6. L56: Please remove hyphen in 'turf-algae' to be consistent with spelling throughout text

7. L57-58: After this sentence, a brief description of the structure of turf algae mats would be instructive to the general reader who may not be familiar with this functional group (e.g. morphology, matrix-like mat of algal branches and filaments and associated bound sediment, height and pervasiveness, etc.)

8. L59-60: As well as favour growth and proliferation of turf algae.

9. L60-61: Turf algae can still provide a highly complex habitat structure, just on a fundamentally smaller scale compared to the architecture provided by larger foundation species such as kelp and corals. Please revise.

10. L74-75: Although feedback loops can make recovery exceedingly more difficult.

11. L77: 'presence and relative strength' rather than 'relative strength and presence'.

12. L93-94, L163-167, L206-207, L249-250: Shifts in competitive balance between foundation species and turf algae are often invoked as underlying this type of regime shift. But be wary that changes in relative abundance along some physical or biological gradient may be driven by other processes.

That is, numerical dominance does not necessarily imply competitive dominance. As the authors have pointed out, CO₂ can be both a resource and a stressor to marine organisms. If one competitor is disproportionately hindered (say calcifying corals), than changes in relative abundance could occur along a gradient of pCO₂ without invoking competition. While competitive processes may be at play here (dense mats of turf algae have been shown elsewhere to inhibit early life-history stages of macroalgae and affect growth and tissue mortality of coral colonies), without a controlled experiment, it is impossible to separate competitive effects from the effects of other stressors and disturbance. Please revise text to provide a more balanced interpretation.

13. L93-94: This is a bit too vague for me and does not properly orient the reader as to what the study specifically sets out to accomplish or how it hopes to advance the field. Please provide some specific questions, objectives, or hypotheses/predictions.

14. Fig 1 Caption: Would be useful to identify the predominant species within the turf algae assemblage.

15. Fig 2 Caption: Should be "Black solid line highlights a continuous shift (Forward threshold = Reverse threshold) from the 'Desirable Ecosystem' with high diversity and complexity, to the 'Degraded Ecosystem' with low diversity and complexity. The black dashed line highlights a discontinuous shift (Forward threshold \neq Reverse threshold) where feedback mechanisms can hinder ecosystem recovery ('hysteresis')."

16. L111: Is there a particular reason why pH was measured on the NBS scale as opposed to a scale more appropriate for seawater? My concern is how comparable the results will be with similar studies. Also, start new sentence with 'Subsequently, ...'

17. There is some pretty dramatic threshold-like behavior in these relationships that readers will find interesting, but which you don't really discuss. I think that itself is a key result that should be described and discussed.

18. Fig. 3 caption: please provide sample sizes for turf, macroalgae, and coral GLM relationships.

19. L116-123: It would also be good to indicate that there were no significant differences in pH, DO, and aragonite saturation between the surface of the turf mat and 2-3 above.

20. Fig. 4: Do dotted lines indicate statistically significant groups from post-hoc comparisons? If so, would be good to indicate in figure caption.
21. L130-132: Please provide error around mean pH values described in text where they correspond with a specific point along the turf depth gradient.
22. Fig. 5a: I think it would be more logical to show dark-adapted measurements with black dots and light-adapted with light grey.
23. Fig. 5a & c: It is curious that water within the sediment layer differed by an average of ~ 0.5 pH units between light- and dark-adapted samples (5a), and yet over the 3-hr light-dark cycles very little change was registered. The magnitude of the difference between steady-state light and dark pH at the turf surface over 3-hr light dark cycles corresponds more closely with the difference between light and dark measurements at the point of maximal pH in 5a. Might fluctuations in pH within the sediment layer occur over a longer, say diel, light-dark cycle? It is difficult to make this evaluation as it is unclear from the methodology whether light and dark measurements were taken within the same turf microcosms over a 12:12 light-dark cycle or on different microcosms adapted either to steady-state light or dark conditions over the 1 week acclimation. It would be helpful to clarify this in the methods sections as well as address the discrepancy between Fig 5 a & c in the Discussion.
24. Fig. 5 caption: It would be good to indicate in the caption that the point of maximal pH (i.e. location of measurements in 5b) is in the upper layer of turf algae.
25. Fig. 6 caption: Should read 'The nine most abundant phyla ...'.
26. L149-154: I think it is misleading to frame these results as effects of turf on recruitment of 'other sessile organisms'. Comparison of the mean cover of crustose coralline algae (CCA) in each treatment described in text with the bars in Fig. 7a would suggest that what the authors are really measuring is recruitment of CCA. If there are data on recruitment of corals, macroalgae, and other sessile taxa they should be presented separately. The story is framed as a feedback loop in which coral and macroalgal recruitment is suppressed in turf-dominated habitats in acidified areas. However, if there is very little (if any) coral and macroalgal recruitment even on tiles without turf, there are likely propagule supply constraints at play given that corals and other macroalgae are very sparse or absent within elevated pCO₂ areas. Therefore, unless there are patches of dense coral and macroalgae in close proximity to the recruitment tiles, it may be impossible with this type of natural experiment to separate recruitment limitations imposed by turf mats inhibiting propagule establishment from recruitment limitations imposed by low propagule supply.
27. L151 & L154: Please specifically reference Fig 7a.
28. L161: Ref 15 would also be appropriate for temperate reefs in addition to Ref 38.
29. L183: I would consider light as a characteristic of the physical environment rather than the chemical microenvironment.
30. L185-188: I think the language should be adjusted here so as not to overstate the results. While the altered physicochemical conditions imposed by expansive turf algae demonstrated by this study could reasonably be speculated to inhibit recruitment of foundation species (coral, large macroalgae) and provide a mechanism for a hysteresis effect, the data in this study does not explicitly demonstrate recruitment inhibition of coral and macroalgae by turf; only the recruitment of CCA (a more favourable settlement substrate for coral and macroalgae). Furthermore, demonstration of hysteresis would require evidence of different thresholds (of pCO₂ in this case) for the forward and reverse shift.
31. L193: Please change to '..., likely fueled by the supply ...'
32. L197: Please change to '... microbial community dominated by sulfur-reducing ...'
33. L199-200: Please change to '... likely contribute to pH decreases, as well as the accumulation of hydrogen sulfide (H₂S) indicated by the characteristic black colouration and smell of these sediments ...' and add comma before 'which'.

34. L206-207: Again, adjust language to distinguish inferences from concrete evidence.
35. L212-216: And spatially variable as evidenced by the gradients in pH, DO, aragonite saturation from turf algae surface to sediment below.
36. L209-228: It seems that the point you are trying to make here is that novel conditions created by turf algae facilitate different types of organisms. That is, the new habitat structure also may drive changes to the trophic structure. But it's not exactly clear. Consider editing this paragraph to clarify this point. Also, would be good to reference Dijkstra et al. 2017 for an example where turf algae facilitate and enhance diversity of small invertebrates compared to larger macroalgae.
37. L230-247: This paragraph addresses how the results of this study and similar ones can inform ecosystem management of coastal marine habitats. However, as it is written, it is still a bit fuzzy for me whether the authors are proposing options available to managers for reversing a regime shift that has already occurred or preventing/reversing a shift that could possibly occur in the near to mid future. I think it is just a matter of some minor editing here. The results of this study from laboratory microcosms simulating end-of-century conditions and CO₂ seep sites (a kind of natural laboratory) provide a window into potential future ecosystem states. Since there is little utility in attempting to mitigate turf abundance at CO₂ seeps, unless there are sites in the region that have shifted to turf habitats as a result of gradual acidification that has already occurred, I think what we're talking about here are options to mitigate future degradation. So just be clear about that.
38. L232-233: Please change to 'positive feedback mechanisms' and 'stabilizing feedback mechanisms'.
39. L240: Should be 'suggest' rather than 'suggests'.
40. L241-242: Are you proposing enhancing resilience of coral/macroalgal habitats by ameliorating locally-manageable human impacts to prevent harmful synergisms with global stressors like acidification that are less manageable at a local level? If so, a reference to Falkenberg et al. (2013) would be appropriate to support that point.
41. L243-244: I think it's reasonable to say that stabilizing feedbacks that lock the turf system in place would likely necessitate more active measures to mitigate turf abundance and facilitate recovery once a regime shift has occurred. But is that a scalable solution? Building resilience of intact desirable ecosystems (previously mentioned) or identifying early warning indicators that signal the approach of an abrupt tipping point may be more achievable.
42. L250: Should be 'key functional groups'.
43. L254: Please delete 'different'.
44. L255-256: Please change to 'The ability of management to reverse an ecological regime shift depends upon an understanding of how species interactions stabilise marine ecosystems.'
45. L266-267: What are the dominant species of coral or macroalgae? A brief description here or in the supplementary material (as you have done for turf assemblage) would be nice.
46. L284-286: I would try modelling these relationships using beta regression and comparing the fit with the negative binomial with AIC as well as looking at the diagnostic plots. It looks like the glm for coral cover is under-predicting coral cover at higher pH values. Beta regression can be used for continuous data falling on the unit interval like proportions. If the fit is improved, you may want to go that route. See <https://cran.r-project.org/web/packages/betareg/vignettes/betareg.pdf>
47. L307 & 377: Please provide the statistic used for global tests and post-hoc comparisons (ANOVA and Tukey HSD). Also, did data meet assumptions for these tests?
48. L324-326: There is no description of profile measurements in dark vs light. Please provide that here. Were dark and light profiles taken on the same microcosms or different sets of replicates for each?
49. L331-332: Microsensor measurements were made under static conditions of no water flow. However, flow has a significant positive effect on the photosynthetic rate of algal turfs (Carpenter &

Williams 2007) and mediates the outcome of interactions between corals, turf, and trapped sediments (Gowan et al. 2014, Jorissen et al. 2016). Given zero flow is an unrealistic assumption under field settings, please acknowledge this caveat and discuss what this means for your lab results in a real world context.

50. L341: Really n = 8. Eight samples each from the turf surface, middle, and sediment layer, respectively.

51. L373-375: As per my previous comment, it's unclear from the text which species or functional groups these 'recruits' refer to. The paper is framed as a turf/macroalgae/coral story, but it seems from the results that what was measured on the PVC tiles was predominantly crustose coralline algae. If any other groups or species were measured please describe that here and in the results. Otherwise, please adjust text to improve accuracy.

52. L370-377: It would be diligent here to provide some description and statistical test (1-way ANOVA) of differences in turf cover between 'treatments'. Since this was more of a natural experiment relying on natural variation in turf covers over the tiles rather than a purposeful manipulation of turf abundance, the authors should demonstrate for the reader that the qualitative treatment names correspond with real quantitative differences.

53. Supplementary Material, para1, L17: should be 'long-term' rather than 'long-time'
Literature Cited

Barott et al. (2009) Hyperspectral and Physiological Analyses of Coral-Algal Interactions. PLoS ONE 4(11):e8043. doi:10.1371/journal.pone.0008043

Barott et al. (2012) Microbial to reef scale interactions between the reef-building coral *Montastraea annularis* and benthic algae. Proc R Soc B 279:1655-1664

Birrell et al. (2005) Effects of algal turfs and sediment on coral settlement. Marine Pollution --Bulletin 51:408-414

Carpenter RC, Williams SL (2007) Mass transfer limitation of photosynthesis of coral reef algal turfs. Marine Biology 151:435-450

Connell SD & Russell BD (2010) The direct effects of increasing CO₂ and temperature on non-calcifying organisms: increasing the potential for phase shifts in kelp forests. Proc R Soc B 277: 1409-1415

Connell et al. (2013) The other ocean acidification problem: CO₂ as a resource among competitors for ecosystem dominance. Phil Trans R Soc B 368: 20120442.

<http://dx.doi.org/10.1098/rstb.2012.0442>

Dijkstra et al. (2017) Invasive seaweeds transform habitat structure and increase biodiversity of associated species. Journal of Ecology 105:1668-1678

Fabricius et al. (2011) Losers and winners in coral reefs acclimatized to elevated carbon dioxide levels. Nature Climate Change 1:165-169

Falkenberg et al. (2013) Disrupting the effects of synergies between stressors: improved water quality dampens the effects of future CO₂ on a marine habitat. Journal of Applied Ecology 50:51-58

Gowan et al. (2014) The effects of water flow and sedimentation on interactions between massive *Porites* and algal turf. Coral Reefs 33:651-663

Jorissen et al. (2016) Evidence for water-mediated mechanisms in coral-algal interactions. Proc R Soc B 283: 20161137.

<http://dx.doi.org/10.1098/rspb.2016.1137>

O'Brien JM & Scheibling RE (2018) Turf wars: competition between foundation and turf-forming species on temperate and tropical reefs and its role in regime shifts. Marine Ecology Progress Series 590:1-17

Smith et al. (2006) Indirect effects of algae on coral: algae-mediated, microbe-induced coral mortality. *Ecology Letters* 9:835-845

Wangpraseurt et al. (2012) In Situ Oxygen Dynamics in Coral-Algal Interactions. *PLoS ONE* 7(2): e31192. doi:10.1371/journal.pone.0031192

Reviewer #1 (Remarks to the Author):

The manuscript 'Feedback mechanisms lock degraded turf algal systems in place' by Harvey et al. focuses on how coastal ecosystem (coral reefs are hereby used as the main example) are subjected to feedback loops during ecosystem degradation.

*The study sums up most of the existing work on feedback mechanisms (I think Barott and Rohwers DDAM model is missing) and presents the main points in a nice and concise form.
Modified as requested to include the work by Barott and Rohwers DDAM model.*

The authors further corroborate the information drawn from the literature with their own findings from a study conducted at CO2 seeps at Shikine Island, Japan. All together this is a solid study that covers the existing literature and the experiments are straight forward.

There are some minor points I would like to have addressed (see below), and one main issue:

These 2 sentences are in my opinion the core of the work: "The stabilising mechanisms maintaining the turf-dominated state here suggests that this ecological shift may exhibit hysteresis, and be more difficult to reverse. Management will need to either promote the resistance and resilience of the existing habitats (e.g. reducing local human pressure), or manipulate turf assemblages to promote recovery of the desirable state (i.e. coral and macroalgae dominated) following the transition"

My personal opinion here is that this is not necessarily groundbreaking new information. What I would like to see is some sort of (even if it is only speculative) more tangible advice or suggestions on how to improve protection or restauration strategies. The authors have shown that the feedback loops play a significant role – what are the implications of this information? What would be some hypothetical actions (based on their findings) that could break these feedback loops? If you would have to restore a degraded reef, how would you go about it, knowing about these feedback loops? I know this is not easy to include, because you have not done the restoration part and it is all just hypothetical. But (my opinion, the editors might have a very different view) if this work should be published in a journal of this quality I think it is important to provide a thought-provoking impulse to make it interesting for a wider audience.

We have now included a paragraph within the discussion that entirely focusses on the potential activities and considerations that would be required by management. This highlights the distinction between reversing an existing shift, and the proactive mitigation of future degradation. We state as follows: 'For ecosystem management, two overarching strategies exist when considering ecosystem shifts; reversing an ecological shift by returning the environmental conditions to prior conditions, or mitigating the future degradation of ecosystems. The issue with attempting to reverse an ecological shift is that whereas a localised impact (such as eutrophication) is reversible through appropriate management, future climate change projections (e.g. ⁶³) would suggest that for global impacts (such as ocean acidification) reversal on timescales relevant to humans is unlikely ²³. Our case study here represents a window into a potential future ecosystem state where an ecosystem has already greatly exceeded its critical threshold. The observed stabilising feedbacks that lock the turf system in place would likely necessitate more active measures to manipulate turf algal mat and facilitate recovery once a regime shift has occurred. While feasible, such an approach is unlikely to present a particularly scalable solution. We suggest that it is more tangible to mitigate future degradation by promoting the resilience of intact desirable ecosystems. One such approach of enhancing the resilience of coral/macroalgal habitats is to ameliorate locally-manageable human impacts (e.g. fisheries, eutrophication) in order to prevent harmful synergisms with global stressors like acidification that are less manageable at a local level ⁶⁴. Ecosystem resilience building may also be achievable through augmentative biocontrol, for example, with sea urchins previously trialed as biocontrol agents of turf algae on coral reefs ^{65,66}, or via the manual removal of undesirable macroalgae ⁶⁷ (the spatial scalability of such solutions is still being assessed). Understanding when management actions are required will be key; foreseeing ecological shifts will require the identification of early warning

indicators that signal the approach of an abrupt tipping point. Continued degradation of ecosystems highlights a need for proactive management, which must incorporate the dynamics of ecosystems in order to mitigate against future degradation.’

Some minor points:

I think the image the authors use in figure 1b depicts the brown algae hydroclathrus. I would not necessarily call it a turf algae. (Turf algae (or “algal turfs”) are dense, multi-species assemblages of filamentous benthic algae, including small individuals of macroalgae and cyanobacteria.)

The predominant species forming the turf algae assemblage in Figure 1b is actually *Biddulphia biddulphiana* (J.E.Smith) Boyer, which forms a thick and dense turf mat across the substratum (in places up to 10 cm thick, as shown in Figure 1B) where it traps large amounts of sediment (~1 kg m⁻²) an example of this is provided in Figure S3). The species identity and description of the turf algal assemblage was previously provided in the supplementary information. Following a comment from Reviewer 2, the name of the predominant species is now listed in the Figure 1 caption, and we additionally now state that “for further description of the turf algal bed, see the supplementary information and ref³⁴).

Methods, microbial sampling. Samples were collected from the middle and the surface of the turf algae. This is somehow vague, I assume a tissue sample was taken from the upper part of a turf algae mat or consortium and one from within one of the turf algal mats, correct? That would make sense, but it is not stated clearly because now I read it as if it was taken from the algal surface and within an alga. But I know that is not what is meant...

You are correct that a sample was taken from the upper part of the turf algae mat, and another sample was taken from within the turf algal mat. We have now re-worded this to improve the clarity. “To characterise the composition of microbial communities associated with turf algal mats, samples were collected by hand from: (i) turf algae located in the upper part of the turf algal mat, (ii) turf algae from within the turf algal mat, and (iii) the sediment below the turf algal mat, from eight haphazard locations within an elevated pCO₂ area (24 samples in total)”

Over all I think this manuscript deserves publication, but I would urge the authors to work on the discussion a bit as described above and be a bit more innovative with their conclusions and recommendations

Thank you for your kind review, we have hopefully now provided a sufficiently improved discussion that more readily provides some tangible and innovative recommendations for management.

Reviewer #2 (Remarks to the Author):

In this study, Harvey and colleagues provide a detailed and mechanistic insight into the processes that could stabilize regime shifts (large-scale and persistent ecosystem reorganizations) in a subtropical marine ecosystem. They find a continuous change in habitat structure along a gradient in pCO₂ at a natural CO₂ seep site in which hard corals and larger macroalgae that enhance 3-dimensional habitat structure are replaced by expansive mats of low-lying turf-forming algae. This provides a window into the type of simplified communities that can be expected to persist with ocean acidification under a 'business-as-usual' climate scenario. From field sampling and laboratory microcosms, they find that seawater carbonate chemistry and dissolved oxygen are altered within the layer of turf algae and accumulated sediments compared to the algal surface and overlying seawater. They also document a gradient in microbial community structure from the surface of the turf algae, to the middle, to the sediment below, which may be responsible for changes to the chemical microenvironment. These microbial-induced chemical changes may make conditions unsuitable for the settlement and survival of other sessile organisms and form an important feedback mechanism that locks the turf ecosystem state in place. This study also found that settlement of crustose coralline algae on PVC tiles, a favourable substrate for settlement of corals and macroalgae, was reduced under dense covers of turf algae.

The study is overall well done and nicely balanced, combining evidence from multiple sources (field sampling, laboratory, water chemistry, genetics) to support a well-rounded and cohesive narrative. The field sampling and laboratory components were appropriately designed and executed. Figures are neatly presented and attractive. Analyses were done properly. The quality of the writing is generally good and clear, but requires more focus and more nuanced interpretation in areas. My main criticism is that I don't think the authors have framed the paper to highlight the novel aspects of the study as well as they could. Other studies have already documented decreased abundance of foundation species like corals and enhanced productivity of turf algae along pCO₂ gradients at seep sites (e.g. Fabricius et al. 2011, Connell et al. 2013). Previous studies have also shown indirect effects of turfs on seawater chemistry (e.g. dissolved oxygen) induced by microbial respiration (Smith et al. 2006, Barott et al. 2009, Wangpraseurt et al. 2012, Jorissen et al. 2016) and shifts in the microbial community at coral-turf interaction zones (Barott et al. 2012). Furthermore, the negative effects of turf algae on recruitment of foundation species are well documented (e.g. corals: Birrell et al. 2005, kelp: Connell & Russell 2010, see O'Brien & Scheibling 2018 for meta-analysis of competition experiments). However, this study is the first to my knowledge to document effects of turf algae on pH and aragonite saturation state and improves on these previous studies by identifying a gradient in carbonate chemistry and microbial communities from the surface of the turf down through the layer of bound sediments. I'm just not sure it's enough to warrant publication in this forum. A general ecological journal may be better suited. I applaud Harvey and co-authors for the amount and quality of work they have accomplished. I think this will make an important contribution to the fields of ecology and conservation. It will be of wide interest to researchers and managers working in diverse marine systems from temperate and tropical regions alike where persistent replacement of foundation species by turf communities (and associated loss of ecosystem services) are becoming increasingly more commonplace. I would be happy to read a revised version of this manuscript whether resubmitted here or another journal.

Thank you for your kind and thorough review, we have now modified the manuscript as requested in order to improve our clarity.

1. L1: Title is a bit too general in my view. Consider revising to convey the main outcome(s) of the study and better inform the reader.

The title has now been changed to 'Feedback mechanisms stabilise degraded turf algal systems at a CO₂ seep site'.

2. L28: 'increase turf algae' is a bit vague. Perhaps 'favour turf algae growth' or 'increase turf algae abundance'?

Modified as requested to 'favour turf algae growth'.

3. L31: The results of this study do not explicitly demonstrate competition for light and space and increased sediment load. Please revise.

Modified as requested to 'altered physicochemical environment and microbial community, and an inhibition of recruitment'.

4. L32-34: Consider revising. Feedback loops can maintain, stabilize, or accelerate degraded shifts to ecosystem states, but often are disconnected from the local to global scale anthropogenic stressors that initially pushed the ecosystem passed a tipping point. Perhaps focus more on the persistence aspect.

Modified as requested to 'Such feedbacks help explain why degraded coastal habitats persist after being initially pushed past the tipping point by global and local anthropogenic stressors.'

5. L52: Should be 'habitats are predominantly composed of species' or 'habitats predominantly comprise'

Modified as requested to 'habitats are predominantly composed of species'.

6. L56: Please remove hyphen in 'turf-algae' to be consistent with spelling throughout text

Modified as requested.

7. L57-58: After this sentence, a brief description of the structure of turf algae mats would be instructive to the general reader who may not be familiar with this functional group (e.g. morphology, matrix-like mat of algal branches an filaments and associated bound sediment, height and pervasiveness, etc.)

Modified as requested. We have now provide a brief description here of turf algal mats in general here, and a more specific description of our mat in the supplementary information.

In the introduction, we state 'These multi-species assemblages of filamentous benthic algae can form dense mats or carpets across extensive areas that can trap large amounts of sediment ¹³'.

In the supplementary information, we state 'The turf algae is a tightly packed matrix-like mat of algal filaments (~5-10 cm in height) that extensively covers the substrate, trapping a thick layer of sediment (predominantly sand, ~3-5 cm deep).'

8. L59-60: As well as favour growth and proliferation of turf algae.

Modified as requested.

9. L60-61: Turf algae can still provide a highly complex habitat structure, just on a fundamentally smaller scale compared to the architecture provided by larger foundation species such as kelp and corals. Please revise.

Modified as requested to 'Finally, although turf algae can provide a highly complex habitat structure, it is on a fundamentally smaller scale compared to that provided by coral reefs and kelp forests ¹⁶ and so the shift to turf algae results in the loss of ecosystem services ¹⁷'.

10. L74-75: Although feedback loops can make recovery exceedingly more difficult.

Modified as requested.

11. L77: 'presence and relative strength' rather than 'relative strength and presence'.

Modified as requested.

12. L93-94, L163-167, L206-207, L249-250: Shifts in competitive balance between foundation species and turf algae are often invoked as underlying this type of regime shift. But be wary that changes in relative abundance along some physical or biological gradient may be driven by other processes. That is, numerical dominance does not necessarily imply competitive dominance. As the authors have pointed out, CO₂ can be both a resource and a stressor to marine organisms. If one competitor is disproportionately hindered (say calcifying corals), than changes in relative abundance could occur along a gradient of pCO₂ without invoking competition. While competitive processes may be at play here (dense mats of turf algae have been shown elsewhere to inhibit early life-history stages of macroalgae and affect growth and tissue mortality of coral colonies), without a controlled experiment, it is impossible to separate competitive effects from the effects of other stressors and disturbance. Please revise text to provide a more balanced interpretation.

Modified as requested.

L93-94 has now been altered following our response to Reviewer Comment 13 (below).

L163-167 now states 'Here, we found that as seawater acidification caused the decline of corals and large macroalgae, there was a major increase in turf algae abundance, indicating that they are capable of monopolising the available space and becoming the most abundant habitat-forming species, corroborating previous works^{28,34}'.

L206-207 now states 'The novel conditions created by the turf algal mat are likely to alter which species will be facilitated and may result in changes to the trophic structure.'

L249-250 now states 'In conclusion, our findings show that seawater acidification changes the relative abundance of key functional groups in marine ecosystems with turf algae being favoured, and that reinforcing feedback loops play an important role in maintaining systems dominated by turf algae.'

13. L93-94: This is a bit too vague for me and does not properly orient the reader as to what the study specifically sets out to accomplish or how it hopes to advance the field. Please provide some specific questions, objectives, or hypotheses/predictions.

Modified as requested. The section now better describes the study, outlining the expectations and predictions.

We state 'Here, we support previous observations that CO₂ enrichment of coastal seawater favours turf algae over corals and macroalgae, and we examine whether this turf algal state is being held in place by stabilising feedback mechanisms and hysteresis. We expected that the presence of turf algae would alter the physicochemical environment within the turf algal habitat. We therefore examine the spatial variability of the carbonate chemistry and dissolved oxygen, and the diurnal variability of pH. Changes in the environmental conditions can lead to (and be affected by) changes in the associated microbial community, and so we characterise the spatial variability in the microbial community composition within the turf algal habitat by DNA metabarcoding using the 16S rRNA gene. Taken together, we predict that these changes in the physicochemical environment and microbial community may cause an inhibition in the recruitment of other macroalgal species and act as a feedback loop that reinforces the persistence of the turf algae. The presence of feedback loops is a necessary condition for the emergence of hysteresis. By transplanting turf algae back along the pCO₂ gradient we assess for differences in the forward and reverse (pCO₂) threshold of turf algae in order to examine whether our system exhibits hysteresis.'

14. Fig 1 Caption: Would be useful to identify the predominant species within the turf algae assemblage.

Modified as requested. The figure caption now includes the identification of the predominant species. We state ‘The predominant species forming the turf algae bed is *Biddulphia biddulphiana* (J.E.Smith) Boyer (for further description of the turf algal bed, see the supplementary information and ref³⁴).’

15. Fig 2 Caption: Should be “Black solid line highlights a continuous shift (Forward threshold = Reverse threshold) from the ‘Desirable Ecosystem’ with high diversity and complexity, to the ‘Degraded Ecosystem’ with low diversity and complexity. The black dashed line highlights a discontinuous shift (Forward threshold \neq Reverse threshold) where feedback mechanisms can hinder ecosystem recovery (‘hysteresis’).”

Modified as requested.

16. L111: Is there a particular reason why pH was measured on the NBS scale as opposed to a scale more appropriate for seawater? My concern is how comparable the results will be with similar studies.

We agree with the reviewer that the NBS scale is not the most appropriate for seawater. At the time of the study, however, we lacked the necessary TRIS buffers to calibrate our pH probes onto the total scale (we have since changed to total scale for our research). It is unfortunately not possible to convert between pH_{NBS} and pH_{T} .

Also, start new sentence with ‘Subsequently,’

Modified as requested.

17. There is some pretty dramatic threshold-like behavior in these relationships that readers will find interesting, but which you don’t really discuss. I think that itself is a key result that should be described and discussed.

We have now included a sentence within the results ‘Overall, the change in abundance of each of the three groups demonstrated relatively abrupt (rather than gradual) changes over the pH gradient suggesting the presence of tipping points / thresholds (Fig. 3).’ and furthermore, we have included this aspect into the discussion as well stating ‘These changes in the abundance of the corals, macroalgae and turf algae along the pCO_2 gradient were abrupt rather than gradual shifts, suggesting that the environmental conditions exceeded a critical threshold⁴³ in the opening paragraph. We further discuss this aspect within the fifth paragraph, stating ‘The transplant of turf algae from the elevated pCO_2 area back along the pCO_2 gradient highlighted that the forward threshold (as shown by the abrupt increase in the natural abundance of the turf habitat as pCO_2 increases) differed from the reverse threshold (as shown by the persistence of the transplanted turf algae under moderate CO_2 enrichment where turf algal habitat does not naturally develop) of our ecosystem state (Fig. 8). The difference between the forward and reverse threshold, combined with the stabilising mechanisms maintaining the turf-dominated state here, suggests that this ecological shift may exhibit hysteresis, and would therefore be particularly difficult to reverse.’

18. Fig. 3 caption: please provide sample sizes for turf, macroalgae, and coral GLM relationships.

Modified as requested. This information is also provided within the text as part of the statistics.

19. L116-123: It would also be good to indicate that there were no significant differences in pH, DO, and aragonite saturation between the surface of the turf mat and 2-3 above.

Modified as requested. We state ‘There were no significant differences in seawater pH, dissolved oxygen or aragonite saturation state between the surface of the turf algae and 2-3 m above (Dunn test, $p > 0.05$).’

20. Fig. 4: Do dotted lines indicate statistically significant groups from post-hoc comparisons? If so, would be good to indicate in figure caption.

We apologise for the confusion, these lines do not indicate the post-hoc comparisons. These lines were meant to provide a visual indication of the turf algal assemblage, and highlight its extent. We have now modified the figure and changed these dotted lines to a shaded region (for the turf algae and

sediment) and explicitly stated its purpose in the figure caption. We have additionally included letters to indicate the post-hoc comparisons between the locations.

21. L130-132: Please provide error around mean pH values described in text where they correspond with a specific point along the turf depth gradient.

Modified as requested.

22. Fig. 5a: I think it would be more logical to show dark-adapted measurements with black dots and light-adapted with light grey.

Modified as requested.

23. Fig. 5a & c: It is curious that water within the sediment layer differed by an average of ~ 0.5 pH units between light- and dark-adapted samples (5a), and yet over the 3-hr light-dark cycles very little change was registered. The magnitude of the difference between steady-state light and dark pH at the turf surface over 3-hr light dark cycles corresponds more closely with the difference between light and dark measurements at the point of maximal pH in 5a. Might fluctuations in pH within the sediment layer occur over a longer, say diel, light-dark cycle? It is difficult to make this evaluation as it is unclear from the methodology whether light and dark measurements were taken within the same turf microcosms over a 12:12 light-dark cycle or on different microcosms adapted either to steady-state light or dark conditions over the 1 week acclimation. It would be helpful to clarify this in the methods sections as well as address the discrepancy between Fig 5 a & c in the Discussion.

We agree with the reviewer that it is possible that the fluctuations in pH within the sediment layer occur over a longer period, and so we have now included a caveat within the discussion ‘Within the sediment, the variations in pH over three-hour light/dark shift periods were minimal compared to the highly productive surface of the turf (Fig. 5b, c), although it might exhibit greater changes over diurnal periods’.

The methodology has now been rewritten in order to provide greater clarity in the description. It now states ‘Our pH microsensor (Unisense pH-50 microsensor, Aarhus, Denmark) was mounted on a manually operated micromanipulator (MMN-333, Narishige, Tokyo, Japan). Measurements were carried out in the following order. We took a pH profile under light conditions (MT-250W Metal halide lamp, MzOne Co., Osaka, Japan) at 200 μ m increments from 25 mm above the mesocosm turf algae into the simulated turf assemblage. Upon reaching the sediment, the microsensor was left in place for 1-hour, and then the pH within the sediment below the turf algae was measured using 3-hour alternating light dark cycles (with measurements taken every 10 seconds) for 9-hours. Following this, the point of maximal pH (near the surface of the turf algae) was located, and after 1-hour, pH was measured for another set of 3-hour alternating light dark cycles (with measurements taken every 10 seconds) over 9-hours. These 3-hour periods for the light dark cycles were sufficient to reach a stable pH value. Once complete, the turf algal microcosm was dark-adapted for three hours, then a second pH profile was performed under dark conditions in a different part of the turf algal microcosm (at 200 μ m increments from 25 mm above the mesocosm turf algae into the simulated turf habitat). All measurements were performed on the same microcosm, and replicated three times using separate turf algal microcosms. Before every pH profile the microsensor was calibrated to the NBS pH scale using three buffers 4.01, 7.00 and 10.01 (Thermo Scientific, USA). Water flow was stopped during the pH profiles and light dark cycles, meaning that measurements were carried out under static water conditions. Variation in the pH of overlying seawater was assessed by taking a measurement using the microsensor and a pH meter (Orion 4-Star, Thermo Scientific, USA). Variation in pH of overlying water (\pm 0.01 pH_{NBS} units) was minimal during our measurements.’

The discrepancy between the pH profile under dark conditions and the light dark cycles carried out in the sediment may be due to the order of the measurements. The measurement of the pH conditions within the sediment were carried out after the pH profile under light conditions. The pH profile under dark conditions, however, was carried out in a new location (within the same microcosm) on the following day after the light dark cycles had been completed (as well as additionally at least a 3-hour dark acclimation period). Therefore, we believe that our results highlight that pH conditions within

the sediment will experience minimal diurnal variation, but the exact pH value may show some spatial variability.

24. *Fig. 5 caption: It would be good to indicate in the caption that the point of maximal pH (i.e. location of measurements in 5b) is in the upper layer of turf algae.*

Modified as requested.

25. *Fig. 6 caption: Should read 'The nine most abundant phyla ...'.*

Modified as requested.

26. *L149-154: I think it is misleading to frame these results as effects of turf on recruitment of 'other sessile organisms'. Comparison of the mean cover of crustose coralline algae (CCA) in each treatment described in text with the bars in Fig. 7a would suggest that what the authors are really measuring is recruitment of CCA. If there are data on recruitment of corals, macroalgae, and other sessile taxa they should be presented separately. The story is framed as a feedback loop in which coral and macroalgal recruitment is suppressed in turf-dominated habitats in acidified areas. However, if there is very little (if any) coral and macroalgal recruitment even on tiles without turf, there are likely propagule supply constraints at play given that corals and other macroalgae are very sparse or absent within elevated pCO₂ areas. Therefore, unless there are patches of dense coral and macroalgae in close proximity to the recruitment tiles, it may be impossible with this type of natural experiment to separate recruitment limitations imposed by turf mats inhibiting propagule establishment from recruitment limitations imposed by low propagule supply.*

We agree with the point raised by the reviewer and have changed the name of this subsection from 'other sessile organisms' to 'other macro algae'. Although the majority of the recruitment is due to CCA, we also found some recruitment by other macroalgae on the tiles without turf algae (but not on those tiles partially or fully covered by turf algae). We have now highlighted this aspect in the results section 'A number of other small macroalgae recruits were also observed on the recruitment tiles in the absence of the turf algae (< 5 % cover, Fig. 7b), but were not found on any of the tiles that were partially or fully covered by turf algae.'

Furthermore, we also agree with the reviewer that it is difficult using a natural experiment to distinguish between the inhibition of recruitment and the recruitment limitations imposed by low propagule supply. It is highly likely within our site that coral propagule supply in the elevated pCO₂ site is low, because the propagule supply is not particularly high in the reference pCO₂ either (due to this being in warm temperate latitudes with the corals at the leading edge of their distribution). We have now acknowledged this caveat within the discussion 'Given the observed reduction in the abundance of both macroalgae and coral species within the elevated pCO₂ area, it is likely that the potential inhibition of recruitment may be further limited by low propagule supply due to the reduced abundance of adults in the nearby area. Both mechanisms would, however, contribute to the hysteresis maintaining the observed regime shift into a turf-dominated state.'

27. *L151 & L154: Please specifically reference Fig 7a.*

Modified as requested.

28. *L161: Ref 15 would also be appropriate for temperate reefs in addition to Ref 38.*

Modified as requested.

29. *L183: I would consider light as a characteristic of the physical environment rather than the chemical microenvironment.*

Modified as requested.

30. *L185-188: I think the language should be adjusted here so as not to overstate the results. While the altered physicochemical conditions imposed by expansive turf algae demonstrated by this study could reasonably be speculated to inhibit recruitment of foundation species (coral, large macroalgae) and provide a mechanism for a hysteresis effect, the data in this study does not explicitly demonstrate*

recruitment inhibition of coral and macroalgae by turf; only the recruitment of CCA (a more favourable settlement substrate for coral and macroalgae). Furthermore, demonstration of hysteresis would require evidence of different thresholds (of pCO₂ in this case) for the forward and reverse shift.

The language in this section has now been tempered in order to highlight that the mechanism could reasonably result in the inhibition of recruitment, which would contribute to the hysteresis.

Additionally, we have now included an additional experiment/result ('subsection - Testing for differences in the forward and reverse threshold along the pCO₂ gradient') which will better support our demonstration of hysteresis. This had initially planned to be included as part of a different study, but we instead include it here as additional evidence. This uses a transplantation of turf algal assemblage on tiles (established in the elevated pCO₂ conditions) back along the pCO₂ gradient. By measuring the persistence of these turf algal assemblages at the different pCO₂ sites (% Cover), we believe that this should represent that the forward and reverse thresholds differ, thereby supporting our suggestion of hysteresis.

31. L193: Please change to '*..., likely fueled by the supply ...*'

Modified as requested.

32. L197: Please change to '*... microbial community dominated by sulfur-reducing ...*'

Modified as requested.

33. L199-200: Please change to '*... likely contribute to pH decreases, as well as the accumulation of hydrogen sulfide (H₂S) indicated by the characteristic black colouration and smell of these sediments ...*' and add comma before '*which*'.

Modified as requested.

34. L206-207: Again, adjust language to distinguish inferences from concrete evidence.

Modified as requested. The language has been reworded to more clearly distinguish between results and inference.

35. L212-216: And spatially variable as evidenced by the gradients in pH, DO, aragonite saturation from turf algae surface to sediment below.

Included as suggested.

36. L209-228: It seems that the point you are trying to make here is that novel conditions created by turf algae facilitate different types of organisms. That is, the new habitat structure also may drive changes to the trophic structure. But it's not exactly clear. Consider editing this paragraph to clarify this point. Also, would be good to reference Dijkstra et al. 2017 for an example where turf algae facilitate and enhance diversity of small invertebrates compared to larger macroalgae.

Yes, the reviewer is correct – we were indeed trying to highlight the link between the novel conditions and the implications for the trophic structure. The paragraph has been restructured, and the Dijkstra et al. 2017 has been included. We now state ‘

The novel conditions created by the turf algal mat are likely to alter which species will be facilitated and may result in changes to the trophic structure. Previous work has demonstrated that shifts to turf algal habitat can enhance the diversity of small invertebrates being facilitated when compared to habitat provided by larger macroalgae species⁶⁰. Here we found that the conditions within the turf algae varied diurnally, showing high primary productivity near the turf surface during light conditions (increasing ~+0.4 pH_{NBS} units compared to the seawater) but reaching a minimum during the dark conditions (decreasing ~-0.3 pH_{NBS} units relative to the seawater). Within the sediment, the variations in pH over three-hour light/dark shift periods were minimal compared to the highly productive surface of the turf (Fig. 5b, c), although it might exhibit greater changes over diurnal periods. These changes to the chemical microenvironment mean that those organisms residing within the turf algal habitat will experience spatially variable pH, as evidenced by the gradients in pH, DO, aragonite saturation from the turf algal mat surface to the sediment below, as well as daily pH fluctuations that both far

exceed the long-term trend predicted for the open ocean. The most abundant taxon of the mobile invertebrates in the turf algal habitat were tanaid crustaceans (~50 % of total abundance in the high-CO₂ and ~10 % in the reference CO₂, Ref³⁴), which can often be found in anoxic conditions⁶¹, as well as at other naturally acidified systems (Papua New Guinea CO₂ seep; Ref⁶²). This lower species evenness suggests that only the stress-tolerant taxa may be facilitated under elevated pCO₂ conditions²⁶. As turf algae become the most abundant habitat, the change to the microenvironment will have a major impact on habitat structure as well as trophic food webs associated with these species.

37. L230-247: This paragraph addresses how the results of this study and similar ones can inform ecosystem management of coastal marine habitats. However, as it is written, it is still a bit fuzzy for me whether the authors are proposing options available to managers for reversing a regime shift that has already occurred or preventing/reversing a shift that could possibly occur in the near to mid future. I think it is just a matter of some minor editing here. The results of this study from laboratory microcosms simulating end-of-century conditions and CO₂ seep sites (a kind of natural laboratory) provide a window into potential future ecosystem states. Since there is little utility in attempting to mitigate turf abundance at CO₂ seeps, unless there are sites in the region that have shifted to turf habitats as a result of gradual acidification that has already occurred, I think what we're talking about here are options to mitigate future degradation. So just be clear about that.

Modified as requested. In response to reviewer 1, we were requested to highlight some potentially suitable management approaches to this issue. Subsequently, we have now restructured this paragraph into two. The second paragraph incorporates the issues that you have raised.

41. L243-244: I think it's reasonable to say that stabilizing feedbacks that lock the turf system in place would likely necessitate more active measures to mitigate turf abundance and facilitate recovery once a regime shift has occurred. But is that a scalable solution? Building resilience of intact desirable ecosystems (previously mentioned) or identifying early warning indicators that signal the approach of an abrupt tipping point may be more achievable.

Modified as requested. We have now reworded this section to better highlight the feasibility of different approaches.

In response to Comment 37 and 41, we now state: 'For ecosystem management, two overarching strategies exist when considering ecosystem shifts; reversing an ecological shift by returning the environmental conditions to prior conditions, or mitigating the future degradation of ecosystems. The issue with attempting to reverse an ecological shift is that whereas a localised impact (such as eutrophication) is reversible through appropriate management, future climate change projections (e.g.⁶³) would suggest that for global impacts (such as ocean acidification) reversal on timescales relevant to humans is unlikely²³. Our case study here represents a window into a potential future ecosystem state where an ecosystem has already greatly exceeded its critical threshold. The observed stabilising feedbacks that lock the turf system in place would likely necessitate more active measures to manipulate turf algal assemblages and facilitate recovery once a regime shift has occurred. While feasible, such an approach is unlikely to present a particularly scalable solution. We suggest that it is more tangible to mitigate future degradation by promoting the resilience of intact desirable ecosystems. One such approach of enhancing the resilience of coral/macroalgal habitats is to ameliorate locally-manageable human impacts (e.g. fisheries, eutrophication) in order to prevent harmful synergisms with global stressors like acidification that are less manageable at a local level⁶⁴. Ecosystem resilience building may also be achievable through augmentative biocontrol, for example, with sea urchins previously trialed as biocontrol agents of turf algae on coral reefs^{65,66}, or via the manual removal of undesirable macroalgae⁶⁷ (the spatial scalability of such solutions is still being assessed). Understanding when management actions are required will be key; foreseeing ecological shifts will require the identification of early warning indicators that signal the approach of an abrupt tipping point. Continued degradation of ecosystems highlights a need for proactive management, which must incorporate the dynamics of ecosystems in order to mitigate against future degradation.'

38. L232-233: Please change to 'positive feedback mechanisms' and 'stabilizing feedback mechanisms'.

Modified as requested.

39. L240: Should be 'suggest' rather than 'suggests'.

Modified as requested.

40. L241-242: Are you proposing enhancing resilience of coral/macroalgal habitats by ameliorating locally-manageable human impacts to prevent harmful synergisms with global stressors like acidification that are less manageable at a local level? If so, a reference to Falkenberg et al. (2013) would be appropriate to support that point.

Modified as requested. A reference to Falkenberg et al. 2013 has now been included. We now state 'One such approach of enhancing the resilience of coral/macroalgal habitats is to ameliorate locally-manageable human impacts (e.g. fisheries, eutrophication) in order to prevent harmful synergisms with global stressors like acidification that are less manageable at a local level⁶⁴'.

42. L250: Should be 'key functional groups'.

Modified as requested.

43. L254: Please delete 'different'.

Modified as requested.

44. L255-256: Please change to 'The ability of management to reverse an ecological regime shift depends upon an understanding of how species interactions stabilise marine ecosystems.'

Modified as requested.

45. L266-267: What are the dominant species of coral or macroalgae? A brief description here or in the supplementary material (as you have done for turf assemblage) would be nice.

Modified as requested. This has been included into the supplementary material.

46. L284-286: I would try modelling these relationships using beta regression and comparing the fit with the negative binomial with AIC as well as looking at the diagnostic plots. It looks like the glm for coral cover is under-predicting coral cover at higher pH values. Beta regression can be used for continuous data falling on the unit interval like proportions. If the fit is improved, you may want to go that route. See <https://cran.r-project.org/web/packages/betareg/vignettes/betareg.pdf>

We thank the reviewer for this suggestion, we actually tried using beta regression previously but found that negative binomial provided a better fit (as assessed by AIC). Beta regression also requires the data to be transformed as it can't handle true 0s or 1s.

We agree that the coral cover is being under-predicted, but we couldn't find a more suitable GLM that provided a better fit (the relationship suggested by beta regression is near linear).

For turf algae, AIC was: 50.4 for negative binomial and -286.7 for beta regression.

For macroalgae, AIC was: 45.1 for negative binomial and -197.8 for beta regression.

For coral, AIC was: 7.70 for negative binomial and -568.8 for beta regression.

47. L307 & 377: Please provide the statistic used for global tests and post-hoc comparisons (ANOVA and Tukey HSD). Also, did data meet assumptions for these tests?

Modified as requested. For the field measurements of the turf assemblage water chemistry, a Kruskal-Wallis and Dunn-test were used. These non-parametric tests were used because they did not meet the assumptions (predominantly due to the inequality in variance). For the recruitment, the assumptions were met and so an ANOVA and Tukey HSD test were used. These details are listed in the methods section for their respective sections.

48. L324-326: *There is no description of profile measurements in dark vs light. Please provide that here. Were dark and light profiles taken on the same microcosms or different sets of replicates for each?*

Modified as requested. This section has now been amended to more clearly describe the methodology and details. We state: 'Our pH microsensor (Unisense pH-50 microsensor, Aarhus, Denmark) was mounted on a manually operated micromanipulator (MMN-333, Narishige, Tokyo, Japan). Measurements were carried out in the following order. We took a pH profile under light conditions (MT-250W Metal halide lamp, MzOne Co., Osaka, Japan) at 200 μm increments from 25 mm above the mesocosm turf algae into the simulated turf assemblage. Upon reaching the sediment, the microsensor was left in place for 1-hour, and then the pH within the sediment below the turf algae was measured using 3-hour alternating light dark cycles (with measurements taken every 10 seconds) for 9-hours. Following this, the point of maximal pH (near the surface of the turf algae) was located, and after 1-hour, pH was measured for another set of 3-hour alternating light dark cycles (with measurements taken every 10 seconds) over 9-hours. These 3-hour periods for the light dark cycles were sufficient to reach a stable pH value. Once complete, the turf algal microcosm was dark-adapted for three hours, then a second pH profile was performed under dark conditions in a different part of the turf algal microcosm (at 200 μm increments from 25 mm above the mesocosm turf algae into the simulated turf habitat). All measurements were performed on the same microcosm, and replicated three times using separate turf algal microcosms. Before every pH profile the microsensor was calibrated to the NBS pH scale using three buffers 4.01, 7.00 and 10.01 (Thermo Scientific, USA). Water flow was stopped during the pH profiles and light dark cycles, meaning that measurements were carried out under static water conditions. Variation in the pH of overlying seawater was assessed by taking a measurement using the microsensor and a pH meter (Orion 4-Star, Thermo Scientific, USA). Variation in pH of overlying water (± 0.01 pH_{NBS} units) was minimal during our measurements.'

49. L331-332: *Microsensor measurements were made under static conditions of no water flow. However, flow has a significant positive effect on the photosynthetic rate of algal turfs (Carpenter & Williams 2007) and mediates the outcome of interactions between corals, turf, and trapped sediments (Gowan et al. 2014, Jorissen et al. 2016). Given zero flow is an unrealistic assumption under field settings, please acknowledge this caveat and discuss what this means for your lab results in a real world context.*

We agree with the reviewer that high flow can have a positive effect on photosynthetic rates of algal turfs. It is therefore possible that the absence of water movement may have limited the metabolic rates (i.e. photosynthesis/metabolism). The pH values recorded in the laboratory experiment are largely in agreement with the values recorded in the field at similar depths within the turf algal assemblage. The intention of the laboratory measurements was to confirm our field observations under more controlled conditions, and since they demonstrated the same patterns, we believe that they are realistic under field settings. The choice to carry out the experiment under zero flow was also partially a pragmatic choice. When using the pump to create water flow, we found it caused electrical interference with the signal. Subsequently, the water flow did not occur during measurements.

50. L341: *Really n = 8. Eight samples each from the turf surface, middle, and sediment layer, respectively.*

Modified as requested. We have now reworded this to state 24 samples in total.

51. L373-375: *As per my previous comment, it's unclear from the text which species or functional groups these 'recruits' refer to. The paper is framed as a turf/macroalgae/coral story, but it seems from the results that what was measured on the PVC tiles was predominantly crustose coralline algae. If any other groups or species were measured please describe that here and in the results. Otherwise, please adjust text to improve accuracy.*

Modified as requested. We have now adjusted the text to highlight that the recruitment predominantly focused on the crustose coralline algae, although other macroalgal species did recruit onto the tiles with no turf cover (zero cover on the tiles with partial or total turf cover).

52. L370-377: *It would be diligent here to provide some description and statistical test (1-way ANOVA) of differences in turf cover between 'treatments'. Since this was more of a natural experiment relying on natural variation in turf covers over the tiles rather than a purposeful manipulation of turf abundance, the authors should demonstrate for the reader that the qualitative treatment names correspond with real quantitative differences.*

Modified as requested. The percentage cover of the turf algae (\pm SD) for each treatment is now provided, as well as a statistical test (Kruskal-Wallis) showing that they differ.

53. *Supplementary Material, para1, L17: should be 'long-term' rather than 'long-time'*

Modified as requested.

Updated Figures

The updated figures are shown below.

Figure 2. Conceptual images for each site were slightly altered to improve clarity.

Figure 4. Shading was added to better highlight the distinction between the surface, turf algae and sediment, and letters for post-hoc tests are included on the right of each panel.

Figure 5. The colour of the points in (a) were reversed.

Figure 8. Additional figure to provide further evidence for our conclusions.

REVIEWERS' COMMENTS:

Reviewer #1 (Remarks to the Author):

In my opinion, the authors have addressed all the issues that were raised by me and the second reviewer. So from that point I have no further objections or concerns that would prohibit publication.

Reviewer #2 (Remarks to the Author):

I thank Harvey and co-authors for the work they have done to address my comments on the original submission. These changes, in combination with the results of an additional transplant experiment to determine the reverse threshold for the observed regime shift and field measurements of turf algae sediment load, have strengthened the study overall. In my view, these revisions are sufficient to warrant publication, provided they can incorporate the following minor changes. Well done!

1. L29-31: As written, this suggests that ocean acidification is reinforcing feedback loops that favour turf algae. However, as you have shown, it is largely the new habitat conditions mediated by the turf-dominated seascape that is creating stabilizing feedback loops. I suggest modifying this sentence to reflect this distinction.

2. Fig. 4: The figure caption suggests the boxplots show mean \pm SE. But I am thinking this is incorrect. Are these rather mean, interquartile range (box), and quartiles \pm 1.5 x IQR (whiskers)?

3. L327: 'other different' is redundant.

4. L380: Should be 'effects of the turf algae' rather than 'effects of the turf algal'

5. L382: Should be '...Dunn test used for ...' rather than '...Dunn test used as for ...'

6. L184-192, L481-495: I think the testing of the forward and reverse threshold greatly strengthens the overall story, and the results are visually striking. However, I think some statistical comparison of turf cover between locations is due to solidify results. I would say also that a statistical comparison between transplant tiles and natural turf cover in nearby areas is required. But if I understand correctly, there is a 2-year separation between measurement of natural turf cover (2017) and the transplant experiment (2019). In that case, a statistical comparison would be contrived.

Reviewer #1 (Remarks to the Author):

In my opinion, the authors have addressed all the issues that were raised by me and the second reviewer. So from that point I have no further objections or concerns that would prohibit publication.

Many thanks for taking the time to review our manuscript for a second time.

Reviewer #2 (Remarks to the Author):

I thank Harvey and co-authors for the work they have done to address my comments on the original submission. These changes, in combination with the results of an additional transplant experiment to determine the reverse threshold for the observed regime shift and field measurements of turf algae sediment load, have strengthened the study overall. In my view, these revisions are sufficient to warrant publication, provided they can incorporate the following minor changes. Well done!

Thank you, and many thanks for taking the time to review our manuscript for a second time.

1. L29-31: As written, this suggests that ocean acidification is reinforcing feedback loops that favour turf algae. However, as you have shown, it is largely the new habitat conditions mediated by the turf-dominated seascape that is creating stabilizing feedback loops. I suggest modifying this sentence to reflect this distinction.

We have now modified the text as suggested.

2. Fig. 4: The figure caption suggests the boxplots show mean \pm SE. But I am thinking this is incorrect. Are these rather mean, interquartile range (box), and quartiles $\pm 1.5 \times$ IQR (whiskers)?

Yes, this is a mistake on our part. We had initially presented the data as barplots \pm error, but following the journal guidelines we changed them to boxplots during the previous revision. However, we forgot to modify the caption. This has now been corrected.

3. L327: 'other different' is redundant.

Removed as suggested.

4. L380: Should be 'effects of the turf algae' rather than 'effects of the turf algal'

Corrected.

5. L382: Should be '...Dunn test used for ...' rather than '...Dunn test used as for ...'

Corrected.

6. L184-192, L481-495: I think the testing of the forward and reverse threshold greatly strengthens the overall story, and the results are visually striking. However, I think some statistical comparison of turf cover between locations is due to solidify results. I would say also that a statistical comparison between transplant tiles and natural turf cover in nearby areas is required. But if I understand correctly, there is a 2-year separation between measurement of natural turf cover (2017) and the transplant experiment (2019). In that case, a statistical comparison would be contrived.

Many thanks. Yes, you are absolutely correct that we avoided the statistics because we felt that they were not appropriate.

Updated Figures

The updated figures are shown below.

Figure 3. Figure was updated following checklist from *Communications Biology* in order to show error distribution.

Figure 8. Figure was updated following checklist from *Communications Biology* in order to show all raw data.